# BOOSTING THE CERTIFIED ROBUSTNESS OF L-INFINITY DISTANCE NETS

**Bohang Zhang**[1]      **Du Jiang**[1]      **Di He**[1,2]      **Liwei Wang**[1,3]

[1]Key Laboratory of Machine Perception, MOE, School of Artificial Intelligence, Peking University
[2]Microsoft Research      [3]International Center for Machine Learning Research, Peking University
zhangbohang@pku.edu.cn   1800013027@pku.edu.cn
di_he@pku.edu.cn   wanglw@cis.pku.edu.cn

## ABSTRACT

Recently, Zhang et al. (2021) developed a new neural network architecture based on $\ell_\infty$-distance functions, which naturally possesses certified $\ell_\infty$ robustness by its construction. Despite the novel design and theoretical foundation, so far the model only achieved comparable performance to conventional networks. In this paper, we make the following two contributions: (i) We demonstrate that $\ell_\infty$-distance nets enjoy a fundamental advantage in certified robustness over conventional networks (under typical certification approaches); (ii) With an improved training process we are able to significantly boost the certified accuracy of $\ell_\infty$-distance nets. Our training approach largely alleviates the optimization problem that arose in the previous training scheme, in particular, the unexpected large Lipschitz constant due to the use of a crucial trick called $\ell_p$-*relaxation*. The core of our training approach is a novel objective function that combines scaled cross-entropy loss and clipped hinge loss with a decaying mixing coefficient. Experiments show that using the proposed training strategy, the certified accuracy of $\ell_\infty$-distance net can be dramatically improved from 33.30% to 40.06% on CIFAR-10 ($\epsilon = 8/255$), meanwhile outperforming other approaches in this area by a large margin. Our results clearly demonstrate the effectiveness and potential of $\ell_\infty$-distance net for certified robustness. Codes are available at https://github.com/zbh2047/L_inf-dist-net-v2.

## 1 INTRODUCTION

Modern neural networks, while achieving high accuracy on various tasks, are found to be vulnerable to small, adversarially-chosen perturbations of the inputs (Szegedy et al., 2013; Biggio et al., 2013). Given an image $x$ correctly classified by a neural network, there often exists a small adversarial perturbation $\delta$, such that the perturbed image $x + \delta$ looks indistinguishable to $x$, but fools the network to predict an incorrect class with high confidence. Such vulnerability creates security concerns in many real-world applications.

A large body of works has been developed to obtain robust classifiers. One line of works proposed heuristic approaches that are empirically robust to particular attack methods, among which adversarial training is the most successful approach (Goodfellow et al., 2014; Madry et al., 2017; Zhang et al., 2019a). However, a variety of these heuristics have been subsequently broken by stronger and adaptive attacks (Carlini & Wagner, 2017; Athalye et al., 2018; Uesato et al., 2018; Tramer et al., 2020; Croce & Hein, 2020), and there are no formal guarantees whether the resulting model is truly robust. This motivates another line of works that seeks certifiably robust classifiers whose prediction is guaranteed to remain the same under all allowed perturbations. Representatives of this field use convex relaxation (Wong & Kolter, 2018; Mirman et al., 2018; Gowal et al., 2018; Zhang et al., 2020b) or randomized smoothing (Cohen et al., 2019; Salman et al., 2019a; Zhai et al., 2020; Yang et al., 2020a). However, these approaches typically suffer from high computational cost, yet still cannot achieve satisfactory results for commonly used $\ell_\infty$-norm perturbation scenario.

Recently, Zhang et al. (2021) proposed a fundamentally different approach by designing a new network architecture called $\ell_\infty$-distance net, a name coming from its construction that the basic neuron is defined as the $\ell_\infty$-distance function. Using the fact that any $\ell_\infty$-distance net is inherently a 1-Lipschitz mapping, one can easily check whether the prediction is certifiably robust for a given data

point according to the output margin. The whole procedure only requires a forward pass without any additional computation. The authors further showed that the model family has strong expressive power, e.g., a large enough $\ell_\infty$-distance net can approximate any 1-Lipschitz function on a bounded domain. Unfortunately, however, the empirical model performance did not well reflect the theoretical advantages. As shown in Zhang et al. (2021), it is necessary to use a conventional multi-layer perception (MLP)[1] on top of an $\ell_\infty$-distance net backbone to achieve better performance compared to the baseline methods. It makes both the training and the certification procedure complicated. More importantly, it calls into question whether the $\ell_\infty$-distance net is really a better model configuration than conventional architectures in the regime of certified robustness.

In this paper, we give an affirmative answer by showing that $\ell_\infty$-distance net *itself* suffices for good performance and can be well learned using an improved training strategy. We first mathematically prove that under mild assumptions of the dataset, there exists an $\ell_\infty$-distance net with reasonable size *by construction* that achieves perfect certified robustness. This result indicates the strong expressive power of $\ell_\infty$-distance nets in robustness certification, and shows a fundamental advantage over conventional networks under typical certification approaches (which do not possess such expressive power according to Mirman et al. (2021)). However, it seems to contradict the previous empirical observations, suggesting that the model may fail to find an optimal solution and further motivating us to revisit the optimization process designed in Zhang et al. (2021).

Due to the non-smoothness of the $\ell_\infty$-distance function, Zhang et al. (2021) developed several training tricks to overcome the optimization difficulty. A notable trick is called the $\ell_p$-relaxation, in which $\ell_p$-distance neurons are used during optimization to give a smooth approximation of $\ell_\infty$-distance. However, we find that the relaxation on neurons unexpectedly relaxes the Lipschitz constant of the network to an exponentially large value, making the objective function no longer maximize the robust accuracy and leading to sub-optimal solutions.

We develop a novel modification of the objective function to bypass the problem mentioned above. The objective function is a linear combination of a scaled cross-entropy term and a modified clipped hinge term. The cross-entropy loss maximizes the output margin regardless of the model's Lipschitzness and makes optimization sufficient at the early training stage when $p$ is small. The clipped hinge loss then focuses on robustness for correctly classified samples at the late training phase when $p$ approaches infinity. The switch from cross-entropy loss to clipped hinge loss is reflected in the mixing coefficient, which decays to zero as $p$ grows to infinity throughout the training procedure.

Despite its simplicity, our experimental results show significant performance gains on various datasets. In particular, an $\ell_\infty$-distance net backbone can achieve **40.06%** certified robust accuracy on CIFAR-10 ($\epsilon = 8/255$). This goes far beyond the previous results, which achieved 33.30% certified accuracy on CIFAR-10 using the same architecture (Zhang et al., 2021). Besides, it surpasses the relaxation-based certification approaches by at least 5 points (Shi et al., 2021; Lyu et al., 2021), establishing a new state-of-the-art result.

To summarize, both the theoretical finding and empirical results in this paper demonstrate the merit of $\ell_\infty$-distance net for certified robustness. Considering the simplicity of the architecture and training strategy used in this paper, we believe there are still many potentials for future research of $\ell_\infty$-distance nets, and more generally, the class of Lipschitz architectures.

## 2 PRELIMINARY

In this section, we briefly introduce the $\ell_\infty$-distance net and its training strategy. An $\ell_\infty$-distance net is constructed using $\ell_\infty$-distance neurons as the basic component. The $\ell_\infty$-distance neuron $u$ takes vector $\boldsymbol{x}$ as the input and calculates the $\ell_\infty$-norm distance between $\boldsymbol{x}$ and parameter $\boldsymbol{w}$ with a bias term $b$. The neuron can be written as

$$u(\boldsymbol{x}, \{\boldsymbol{w}, b\}) = \|\boldsymbol{x} - \boldsymbol{w}\|_\infty + b. \tag{1}$$

Based on the neuron definition, a fully-connected $\ell_\infty$-distance net can then be constructed. Formally, an $L$ layer network $\boldsymbol{g}$ takes $\boldsymbol{x}^{(0)} = \boldsymbol{x}$ as the input, and the $l$th layer $\boldsymbol{x}^{(l)}$ is calculated by

$$x_i^{(l)} = u(\boldsymbol{x}^{(l-1)}, \{\boldsymbol{w}^{(l,i)}, b_i^{(l)}\}) = \|\boldsymbol{x}^{(l-1)} - \boldsymbol{w}^{(l,i)}\|_\infty + b_i^{(l)}, \quad l \in [L], i \in [n_l]. \tag{2}$$

---

[1]Without any confusion, in this paper, a *conventional* neural network model is referred to as a network composed of linear transformations with non-linear activations.

Here $n_l$ is the number of neurons in the $l$th layer. For $K$-class classification problems, $n_L = K$. The network outputs $\boldsymbol{g}(\boldsymbol{x}) = \boldsymbol{x}^{(L)}$ as logits and predicts the class $\arg\max_{i \in [K]}[\boldsymbol{g}(\boldsymbol{x})]_i$.

An important property of $\ell_\infty$-distance net is its Lipschitz continuity, as is stated below.

**Definition 2.1.** A mapping $\boldsymbol{f}(\boldsymbol{z}) : \mathbb{R}^m \to \mathbb{R}^n$ is called $\lambda$-Lipschitz with respect to $\ell_p$-norm $\|\cdot\|_p$, if for any $\boldsymbol{z}_1, \boldsymbol{z}_2$, the following holds:

$$\|\boldsymbol{f}(\boldsymbol{z}_1) - \boldsymbol{f}(\boldsymbol{z}_2)\|_p \leq \lambda \|\boldsymbol{z}_1 - \boldsymbol{z}_2\|_p. \tag{3}$$

**Proposition 2.2.** *The mapping of an $\ell_\infty$-distance layer is 1-Lipschitz with respect to $\ell_\infty$-norm. Thus by composition, any $\ell_\infty$-distance net $\boldsymbol{g}(\cdot)$ is 1-Lipschitz with respect to $\ell_\infty$-norm.*

$\ell_\infty$-distance nets naturally possess certified robustness using the Lipschitz property. In detail, for any data point $\boldsymbol{x}$ with label $y$, denote the output margin of network $\boldsymbol{g}$ as

$$\mathsf{margin}(\boldsymbol{x}, y; \boldsymbol{g}) = [\boldsymbol{g}(\boldsymbol{x})]_y - \max_{j \neq y}[\boldsymbol{g}(\boldsymbol{x})]_j. \tag{4}$$

If $\boldsymbol{x}$ is correctly classified by $\boldsymbol{g}$, then the prediction of a perturbed input $\boldsymbol{x} + \boldsymbol{\delta}$ will remain the same as $\boldsymbol{x}$ if $\|\boldsymbol{\delta}\|_\infty < \mathsf{margin}(\boldsymbol{x}, y; \boldsymbol{g})/2$. In other words, we can obtain the certified robustness for a given perturbation level $\epsilon$ according to $\mathbb{I}(\mathsf{margin}(\boldsymbol{x}, y; \boldsymbol{g})/2 > \epsilon)$, where $\mathbb{I}(\cdot)$ is the indicator function. We call this *margin-based certification*. Given this certification approach, a corresponding training approach can then be developed, where one simply learns a large margin classifier using standard loss functions, e.g., hinge loss, without resorting to adversarial training. Therefore the whole training procedure is as efficient as training standard networks with *no* additional cost.

Zhang et al. (2021) further show that $\ell_\infty$-distance nets are Lipschitz-universal approximators. In detail, a large enough $\ell_\infty$-distance net can approximate any 1-Lipschitz function with respect to $\ell_\infty$-norm on a bounded domain arbitrarily well.

**Training $\ell_\infty$-distance nets.** One major challenge in training $\ell_\infty$-distance net is that the $\ell_\infty$-distance operation is highly non-smooth, and the gradients (i.e. $\nabla_{\boldsymbol{x}}\|\boldsymbol{x} - \boldsymbol{w}\|_\infty$ and $\nabla_{\boldsymbol{w}}\|\boldsymbol{x} - \boldsymbol{w}\|_\infty$) are sparse. To mitigate the problem, Zhang et al. (2021) used $\ell_p$-distance neurons instead of $\ell_\infty$-distance ones during training, resulting in approximate and non-sparse gradients. Typically $p$ is set to a small value (e.g., 8) in the beginning and increases throughout training until it reaches a large number (e.g., 1000). The authors also designed several other tricks to further address the optimization difficulty. However, even with the help of tricks, $\ell_\infty$-distance nets only perform competitively to previous works. The authors thus considered using a hybrid model architecture, in which the $\ell_\infty$-distance net serves as a robust feature extractor, and an additional conventional multi-layer perceptron is used as the prediction head. This architecture achieves the best performance, but both the training and the certification approach become complicated again due to the presence of non-Lipschitz MLP layers.

## 3 EXPRESSIVE POWER OF $\ell_\infty$-DISTANCE NETS IN ROBUST CLASSIFICATION

In this section, we challenge the conclusion of previous work by proving that simple $\ell_\infty$-distance nets (without the top MLP) suffice for achieving *perfect* certified robustness in classification. Recall that Zhang et al. (2021) already provides a universal approximation theorem, showing the expressive power of $\ell_\infty$-distance nets to represent Lipschitz functions. However, their result focuses on real-valued function approximation and is not directly helpful for certified robustness in classification. One may ask: Does a certifiably robust $\ell_\infty$-distance net exist *given a dataset*? If so, how large does the network need to be? We will answer these questions and show that one can explicitly *construct* an $\ell_\infty$-distance net that achieves perfect certified robustness as long as the dataset satisfies the following (weak) condition called $r$-separation (Yang et al., 2020b).

**Definition 3.1.** ($r$-separation) Consider a labeled dataset $\mathcal{D} = \{(\boldsymbol{x}_i, y_i)\}$ where $y_i \in [K]$ is the label of $\boldsymbol{x}_i$. We say $\mathcal{D}$ is $r$-separated with respect to $\ell_p$-norm if for any pair of samples $(\boldsymbol{x}_i, y_i), (\boldsymbol{x}_j, y_j)$, as long as $y_i \neq y_j$, one has $\|\boldsymbol{x}_i - \boldsymbol{x}_j\|_p > 2r$.

Table 1: The $r$-separation property of commonly used datasets, taken from Yang et al. (2020b).

| Dataset | $r$ | commonly used $\epsilon$ |
|---------|-----|--------------------------|
| MNIST | 0.369 | 0.3 |
| CIFAR-10 | 0.106 | 8/255 |

It is easy to see that $r$-separation is a *necessary* condition for robustness under $\ell_p$-norm perturbation $\epsilon = r$. In fact, the condition holds for all commonly used datasets (e.g., MNIST, CIFAR-10): the value of $r$ in each dataset is much greater than the allowed perturbation level $\epsilon$ as is demonstrated in Yang et al. (2020b) (see Table 1 above). The authors took a further step and showed there must exist a classifier that achieves perfect robust accuracy if the condition holds. We now prove that even if we restrict the classifier to be the network function class represented by $\ell_\infty$-distance nets, the conclusion is still correct: a simple two-layer $\ell_\infty$-distance net with hidden size $O(n)$ can already achieve perfect robustness for $r$-separated datasets.

**Theorem 3.2.** *Let $\mathcal{D}$ be a dataset with $n$ elements satisfying the $r$-separation condition with respect to $\ell_\infty$-norm. Then there exists a two-layer $\ell_\infty$-distance net with hidden size $n$, such that when using margin-based certification, the certified $\ell_\infty$ robust accuracy is 100% on $\mathcal{D}$ under perturbation $\epsilon = r$.*

*Proof sketch.* Consider a two layer $\ell_\infty$-distance net $\boldsymbol{g}$ defined in Equation (2). Let its parameters be assigned by

$$\boldsymbol{w}^{(1,i)} = \boldsymbol{x}_i, b_i^{(1)} = 0 \qquad \text{for } i \in [n]$$
$$w_i^{(2,j)} = C \cdot \mathbb{I}(y_i = j), b_j^{(2)} = -C \qquad \text{for } i \in [n], j \in [K]$$

where $C = 4\max_{i\in[n]} \|\boldsymbol{x}_i\|_\infty$ is a constant, and $\mathbb{I}(\cdot)$ is the indicator function. For the above assignment, it can be proved that the network outputs

$$[\boldsymbol{g}(\boldsymbol{x})]_j = x_j^{(2)} = -\min_{i\in[n],y_i=j} \|\boldsymbol{x} - \boldsymbol{x}_i\|_\infty. \tag{5}$$

From Equation (5) the network $\boldsymbol{g}$ can represent a nearest neighbor classifier, in that it outputs the negative of the nearest neighbor distance between input $\boldsymbol{x}$ and the samples of each class. Therefore, given data $\boldsymbol{x} = \boldsymbol{x}_i$ in dataset $\mathcal{D}$, the output margin of $\boldsymbol{g}(\boldsymbol{x})$ is at least $2r$ due to the $r$-separation condition. In other words, $\boldsymbol{g}$ achieves 100% certified robust accuracy on $\mathcal{D}$. $\qquad\square$

**Remark 3.3.** The above result can be extended to multi-layer networks. In general, we can prove the existence of such networks with $L$ layers and no more than $O(n/L + K + d)$ hidden neurons for each hidden layer where $d$ is the input dimension. See Appendix A for details of the proof.

The significance of Theorem 3.2 can be reflected in the following two aspects. Firstly, our result explicitly shows the strong expressive power of $\ell_\infty$-distance nets in *robust classification*, which complements the universal approximation theorem in Zhang et al. (2021). Moreover, Theorem 3.2 gives an upper bound of $O(n)$ on the required network size which is close to practical applications. It is much smaller than the size needed for function approximation ($O(1/\varepsilon^d)$ under approximation error $\varepsilon$, proved in Zhang et al. (2021)), which scales exponentially in the input dimension $d$.

Secondly, our result justifies that *for well-designed architectures*, using only the *global* Lipschitz property is *sufficient* for robustness certification. It contrasts to the prior view that suggests leveraging the *local* Lipschitz constant is necessary (Huster et al., 2018), which typically needs sophisticated calculations (Wong et al., 2018; Zhang et al., 2018; 2020b). More importantly, as a comparison, Mirman et al. (2021) very recently proved that for any conventional network, the commonly-used *interval bound propagation* (IBP) (Mirman et al., 2018; Gowal et al., 2018) intrinsically cannot achieve perfect certified robustness on a simple $r$-separation dataset containing only three data points (under $\epsilon = r$). In other words, $\ell_\infty$-distance nets certified using the global Lipschitz property have a fundamental advantage over conventional networks certified using interval bound propagation.

## 4 INVESTIGATING THE TRAINING OF $\ell_\infty$-DISTANCE NETS

Since robust $\ell_\infty$-distance nets exist in principle, the remaining thing is understanding why the current training strategy cannot find a robust solution. In this section, we first provide evidence that the training method in Zhang et al. (2021) is indeed problematic and cannot achieve good certified robustness, then provide a novel way to address the issue.

### 4.1 PROBLEMS OF THE CURRENT TRAINING STRATEGY

As shown in Section 2, given any perturbation level $\epsilon$, the certified accuracy of data point $\boldsymbol{x}$ can be calculated according to $\mathbb{I}(\mathsf{margin}(\boldsymbol{x}, y; \boldsymbol{g})/2 > \epsilon)$ for 1-Lipschitz functions. Then the hinge loss becomes standard to learn a robust $\ell_\infty$-distance net:

$$\mathcal{L}_{\text{hinge}}(\boldsymbol{g}, \mathcal{D}; \theta) = \mathbb{E}_{(\boldsymbol{x}_i, y_i)\in\mathcal{D}} \left[\max\{\theta - \mathsf{margin}(\boldsymbol{x}_i, y_i; \boldsymbol{g}), 0\}\right], \tag{6}$$

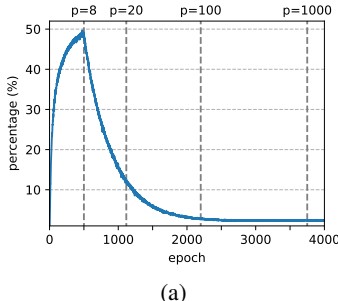

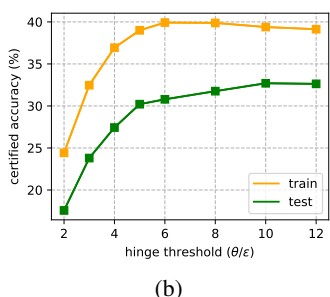

(a)                                                              (b)

Figure 1: Experiments of $\ell_\infty$-distance net training on CIFAR-10 dataset using the hinge loss function. Training details can be found in Section 5.1. (a) The percentage of training samples with output margin greater than $\theta$ throughout training. We use very long training epochs, and the final percentage is still below 2.5%. The dashed lines indicate different $p$ values of $\ell_p$-distance neurons. (b) The certified accuracy on training dataset and test dataset respectively, trained using different hinge threshold $\theta$. Training gets worse when $\theta \leq 6\epsilon$.

where $\theta$ is the hinge threshold which should be larger than $2\epsilon$. Hinge loss aims at making the output margin for any sample greater than $\theta$, and $\mathcal{L}_{\text{hinge}}(\boldsymbol{g}, \mathcal{D}; \theta) = 0$ if and only if the network $\boldsymbol{g}$ achieves perfect certified robustness on training dataset under perturbation $\epsilon = \theta/2$.

**Hinge loss fails to learn robust classifiers.**   We first start with some empirical observations. Consider a plain $\ell_\infty$-distance net trained on CIFAR-10 dataset using the same approach and hyper-parameters provided in Zhang et al. (2021). When the training finishes, we count the percentage of training samples whose margin is greater than $\theta$, i.e., achieving zero loss. We expect the value to be large if the optimization is successful. However, the result surprisingly reveals that only 1.62% of the training samples are classified correctly with a margin greater than $\theta$. Even if we use much longer training epochs (e.g. 4000 epochs on CIFAR-10), the percentage is still below 2.5% (see Figure 1(a)). Thus we conclude that hinge loss fails to optimize well for most of the training samples.

Since the output margins of the vast majority of training samples are less than $\theta$, the loss approximately degenerates to a linear function without the maximum operation:

$$\mathcal{L}_{\text{hinge}}(\boldsymbol{g}, \mathcal{D}; \theta) \approx \widetilde{\mathcal{L}}_{\text{hinge}}(\boldsymbol{g}, \mathcal{D}; \theta) = \mathbb{E}_{(\boldsymbol{x}_i, y_i) \in \mathcal{D}} \left[ \theta - \text{margin}(\boldsymbol{x}_i, y_i; \boldsymbol{g}) \right]$$
$$= \theta - \mathbb{E}_{(\boldsymbol{x}_i, y_i) \in \mathcal{D}} \left[ \text{margin}(\boldsymbol{x}_i, y_i; \boldsymbol{g}) \right], \quad (7)$$

where $\theta$ becomes irrelevant, and training becomes to optimize *the average margin over the dataset* regardless of the allowed perturbation $\epsilon$ which is definitely problematic.

After checking the hyper-parameters used in Zhang et al. (2021), we find that the hinge threshold $\theta$ is set to 80/255, which is *much larger* than the perturbation level $\epsilon = 8/255$. This partly explains the above degeneration phenomenon, but it is still unclear why such a large value has to be taken. We then conduct experiments to see the performance with different chosen hinge thresholds $\theta$. The results are illustrated in Figure 1(b). As one can see, a smaller hinge threshold not only reduces certified test accuracy but even makes *training* worse.

**Why does this happen?**   We find that the reason for the loss degeneration stems from the $\ell_p$-relaxation used in training. While $\ell_p$-relaxation alleviates the sparse gradient problem, it destroys the Lipschitz property of the $\ell_\infty$-distance neuron, as stated in Proposition 4.1:

**Proposition 4.1.** *A layer constructed using $\ell_p$-distance neurons*

$$u_p(\boldsymbol{x}, \{\boldsymbol{w}, b\}) = \|\boldsymbol{x} - \boldsymbol{w}\|_p + b \quad (8)$$

*is $d^{1/p}$ Lipschitz with respect to $\ell_\infty$-norm, where $d$ is the dimension of $\boldsymbol{x}$. Thus by composition, an $L$ layer $\ell_p$-distance net is $d^{L/p}$ Lipschitz with respect to $\ell_\infty$-norm.*

Zhang et al. (2021) uses a small $p$ in the beginning and increases its value gradually during training. From Proposition 4.1, the Lipschitz constant of the smoothed network can be significantly large[2] at

---

[2]For a 6-layer $\ell_p$-distance net ($p = 8$) with hidden size 5120 used in Zhang et al. (2021), Proposition 4.1 approximately gives a Lipschitz constant of 568. We also run experiments to validate such upper bound is relatively tight. See Appendix C for more details.

the early training stage. Note that the robustness certification $\mathbb{I}(\text{margin}(\boldsymbol{x}, y; \boldsymbol{g})/2 \leq \epsilon)$ holds for 1-Lipschitz functions. When the Lipschitz constant is large, even if the margin of a data point passes the threshold, the data point can still lie near to classification boundary. This makes the training using hinge loss ineffective at the early stage and converge to a wrong solution far from the real optima. We argue that the early-stage training is important as when $p$ rises to a large value, the optimization becomes intrinsically difficult to push the parameters back to the correct solution due to sparse gradients.

Such argument can be verified from Figure 1(a), in which we plot the percentage of training samples with margins greater than $\theta$ throughout a long training process. After $p$ starts to rise, the margin decreases drastically, and the percentage sharply drops and never increases again during the whole $\ell_p$-relaxation procedure. It is also clear why the value of $\theta$ must be chosen to be much larger than $2\epsilon$. For small $\theta$, the margin optimization becomes insufficient at early training stages when the Lipschitz constant is exponentially large, leading to worse performance *even on the training dataset*.

## 4.2 OUR SOLUTION

In the previous section, one can see that the hinge loss and the $\ell_p$-relaxation are incompatible. As the $\ell_p$-relaxation is essential to overcome the sparse gradient problem, we focus on developing better loss functions. We show in this section that a simple change of the objective function can address the above problem, leading to non-trivial improvements.

We approach the issue by investigating the commonly used cross-entropy loss. For conventional non-Lipschitz networks, cross-entropy loss aims at increasing the logit of the true class while decreasing the other logits as much as possible, therefore enlarges the output margin without a threshold constraint. This makes the optimization of output margin sufficient regradless of the Lipschitz constant, which largely alleviates the problem in Section 4.1 for training $\ell_p$-distance nets with small $p$. Such findings thus motivate us to replace hinge loss with cross-entropy. However, on the other hand, cross-entropy loss only coarsely enlarges the margin, rather than exactly optimizing the surrogate of certified accuracy $\mathbb{I}(\text{margin}(\boldsymbol{x}, y; \boldsymbol{g})/2 \leq \epsilon)$ that depends on $\epsilon$. When the model is almost 1-Lipschitz (i.e. $p$ approaches infinity), hinge loss can still be better than cross-entropy. In other words, cross-entropy loss and hinge loss are complementary.

Based on the above argument, we propose to simply combine cross-entropy loss and hinge loss to obtain the following objective function:

$$\mathcal{L}(\boldsymbol{g}, \mathcal{D}; \theta) = \mathbb{E}_{(\boldsymbol{x}_i, y_i) \in \mathcal{D}}[\underbrace{\lambda \ell_{\text{CE}}(s \cdot \boldsymbol{g}(\boldsymbol{x}_i), y_i)}_{\text{scaled cross-entropy loss}} + \underbrace{\min\{\ell_{\text{hinge}}(\boldsymbol{g}(\boldsymbol{x}_i)/\theta, y_i), 1\}}_{\text{clipped hinge loss}}] \tag{9}$$

where $\ell_{\text{CE}}(\boldsymbol{z}, y) = \log(\sum_i \exp(z_i)) - z_y$ and $\ell_{\text{hinge}}(\boldsymbol{z}, y) = \max\{\max_{i \neq y} z_i - z_y + 1, 0\}$. We now make detailed explanations about each term in Equation (9).

**Scaled cross-entropy loss** $\ell_{\text{CE}}(s \cdot \boldsymbol{g}(\boldsymbol{x}_i), y_i)$. Cross-entropy loss deals with the optimization issue when $p$ is small. Here a slight difference is the introduced scaling $s$ as is explained below. We know cross-entropy loss is invariant to the shift operation (adding a constant to each output logit) but not scaling (multiplying a constant). For conventional networks, the output logits are produced through the last linear layer, and the scaling factor can be implicitly learned in the parameters of the linear layer to match the cross-entropy loss. However, $\ell_\infty$-distance net is strictly 1-Lipschitz and does not have any scaling operation. We thus introduce a learnable scalar (temperature) $s$ that multiplies the network output $\boldsymbol{g}(\boldsymbol{x}_i)$ before taking cross-entropy loss. We simply initialize it to be 1. Note that $s$ does not influence the classification results and can be removed once the training finishes.

**Clipped hinge loss** $\min\{\ell_{\text{hinge}}(\boldsymbol{g}(\boldsymbol{x}_i)/\theta, y_i), 1\}$. Clipped hinge loss is designed to achieve robustness when $p$ approaches infinity. Unlike the standard hinge loss, the clipped version plateaus if the output margin is negative (i.e., misclassified). In other words, clipped hinge loss is equivalent to applying hinge loss only on correctly-classified samples. The reason for applying such a clipping is three-fold. (i) Scaled cross-entropy loss already focuses on learning a model with high (clean) accuracy, thus there is no need to optimize on wrongly-classified samples duplicatively using hinge loss. Moreover, cross-entropy is better than hinge loss when used in classification, as the gradient of hinge loss makes optimization harder[3]. (ii) In the late training phase, the optimization becomes

---

[3]The gradient of hinge loss w.r.t. output logits is sparse (only two non-zero elements) and does not make full use of the information provided by the logit.

intrinsically difficult ($p$ approaching infinity). As a consequence, wrongly classified samples may have little chance to be robust. Clipped hinge loss thus ignores these samples and concentrates on easier ones to increase their potential to be robust after training. (iii) The clipped hinge loss is a better surrogate for 0-1 robust error $\mathbb{I}(\mathrm{margin}(\boldsymbol{x}, y; \boldsymbol{g})/2 \leq \epsilon)$. Compared with hinge loss, the clipped version is closer to our goal and more likely to achieve better certified accuracy. Finally, due to the presence of cross-entropy loss, we will show in Section 5.3 that the hinge threshold $\theta$ can be set to a much smaller value unlike Figure 1(b), which thus avoids the loss degeneration problem.

**The mixing coefficient** $\lambda$. The coefficient $\lambda$ in loss (9) plays a role in the trade-off between cross-entropy and hinge loss. Based on the above motivation, we use a decaying $\lambda$ that attenuates from $\lambda_0$ to zero throughout the process of $\ell_p$-relaxation ($\lambda_0$ is a hyper-parameter). When $p$ is small at the early training stage, we focus more on cross-entropy loss. After $p$ grows large, $\lambda$ vanishes, and a surrogate of 0-1 robust error is optimized.

We point out that objective functions similar to (9) also appeared in previous literature. In particular, the TRADES loss (Zhang et al., 2019a) and MMA loss (Ding et al., 2020) are both composed of a mixture of the cross-entropy loss and a form of robust loss. Nevertheless, the motivations of these methods are quite different. For example, TRADES was proposed based on the theoretical results suggesting robustness may be at odds with accuracy (Tsipras et al., 2019), while our training approach is mainly motivated by the optimization issue. Furthermore, the implementations of these methods also vary a lot. We use clipped hinge loss to achieve robustness due to its simplicity, and uses a decaying $\lambda$ correlate to $p$ in $\ell_p$-relaxation due to the optimization problem in Section 4.1.

## 5 EXPERIMENTS

### 5.1 EXPERIMENTAL SETTING

In this section, we evaluate the proposed training strategy on benchmark datasets MNIST and CIFAR-10 to show the effectiveness of $\ell_\infty$-distance net.

**Model details.** We use exactly the same model as Zhang et al. (2021) for a fair comparison. Concretely, we consider the simple fully-connect $\ell_\infty$-distance nets defined in Equation (2). All hidden layers have 5120 neurons. We use a 5-layer network for MNIST and a 6-layer one for CIFAR-10.

**Training details.** In all experiments, we choose the Adam optimizer with a batch size of 512. The learning rate is set to 0.03 initially and decayed using a simple cosine annealing throughout the whole training process. We use padding and random crop data augmentation for MNIST and CIFAR-10, and also use random horizontal flip for CIFAR-10. The $\ell_p$-relaxation starts at $p = 8$ and ends at $p = 1000$ with $p$ increasing exponentially. Accordingly, the mixing coefficient $\lambda$ decays exponentially during the $\ell_p$-relaxation process from $\lambda_0$ to a vanishing value $\lambda_{\mathrm{end}}$. We do not use further tricks that are used in Zhang et al. (2021), e.g. the $\ell_p$ weight decay or a warmup over perturbation $\epsilon$, to keep our training strategy clean and simple. The dataset dependent hyper-parameters, including $\theta$, $\lambda_0$, $\lambda_{\mathrm{end}}$ and the number of epochs $T$, can be found in Appendix B. All experiments are run for 8 times on a single NVIDIA Tesla-V100 GPU, and the median of the performance number is reported.

**Evaluation.** We test the robustness of the trained models under $\epsilon$-bounded $\ell_\infty$-norm perturbations. Following the common practice (Madry et al., 2017), we mainly use $\epsilon = 0.3$ for MNIST dataset and $8/255$ for CIFAR-10 dataset. We also provide results under other perturbation magnitudes, e.g. $\epsilon = 0.1$ for MNIST and $\epsilon = 2/255, \epsilon = 16/255$ for CIFAR-10. We first evaluate the robust test accuracy under the Projected Gradient Descent (PGD) attack (Madry et al., 2017). The number of iterations of the PGD attack is set to a large number of 100. We then calculate the certified robust accuracy based on the output margin.

### 5.2 EXPERIMENTAL RESULTS

Results are presented in Table 2. For each method in the table, we report the clean test accuracy without perturbation (denoted as Clean), the robust test accuracy under PGD attack (denoted as PGD), and the certified robust test accuracy (denoted as Certified). We also compare with randomized smoothing (see Appendix D), despite these methods provides probabilistic certified guarantee and usually take thousands of times more time than other approaches for robustness certification.

Table 2: Comparison of our results with existing methods.

| Dataset | $\epsilon$ | Method | Reference | Clean | PGD | Certified |
|---|---|---|---|---|---|---|
| MNIST | 0.1 | CAP | (Wong et al., 2018) | 98.92 | - | 96.33 |
| | | IBP* | (Gowal et al., 2018) | 98.92 | 97.98 | 97.25 |
| | | CROWN-IBP | (Zhang et al., 2020b) | 98.83 | 98.19 | 97.76 |
| | | IBP | (Shi et al., 2021) | 98.84 | - | **97.95** |
| | | COLT | (Balunovic & Vechev, 2020) | 99.2 | - | 97.1$^{\parallel}$ |
| | | $\ell_\infty$-distance Net$^{\dagger}$ | (Zhang et al., 2021) | 98.66 | 97.79$^{\ddagger}$ | 97.70 |
| | | $\ell_\infty$-distance Net | This paper | 98.93 | 98.03 | **97.95** |
| | 0.3 | IBP* | (Gowal et al., 2018) | 97.88 | 93.22 | 91.79 |
| | | CROWN-IBP | (Zhang et al., 2020b) | 98.18 | 93.95 | 92.98 |
| | | IBP | (Shi et al., 2021) | 97.67 | - | **93.10** |
| | | COLT | (Balunovic & Vechev, 2020) | 97.3 | - | 85.7$^{\parallel}$ |
| | | $\ell_\infty$-distance Net+MLP | (Zhang et al., 2021) | 98.56 | 95.28$^{\ddagger}$ | 93.09 |
| | | $\ell_\infty$-distance Net | (Zhang et al., 2021) | 98.54 | 94.71$^{\ddagger}$ | 92.64 |
| | | $\ell_\infty$-distance Net | This paper | 98.56 | 94.73 | **93.20** |
| CIFAR-10 | 2/255 | CAP | (Wong et al., 2018) | 68.28 | - | 53.89 |
| | | IBP* | (Gowal et al., 2018) | 61.46 | 50.28 | 44.79 |
| | | CROWN-IBP | (Zhang et al., 2020b) | 71.52 | 59.72 | 53.97 |
| | | IBP | (Shi et al., 2021) | 66.84 | - | 52.85 |
| | | COLT | (Balunovic & Vechev, 2020) | 78.4 | - | 60.5$^{\parallel}$ |
| | | Randomized Smoothing | (Blum et al., 2020) | 78.8 | - | **62.6**$^{\S\parallel}$ |
| | | $\ell_\infty$-distance Net$^{\dagger}$ | (Zhang et al., 2021) | 60.33 | 51.45$^{\ddagger}$ | 50.94 |
| | | $\ell_\infty$-distance Net | This paper | 60.61 | 54.28 | **54.12** |
| | 8/255 | IBP* | (Gowal et al., 2018) | 50.99 | 31.27 | 29.19 |
| | | CROWN-IBP | (Zhang et al., 2020b) | 45.98 | 34.58 | 33.06 |
| | | CROWN-IBP | (Xu et al., 2020) | 46.29 | 35.69 | 33.38 |
| | | IBP | (Shi et al., 2021) | 48.94 | - | 34.97 |
| | | CROWN-LBP | (Lyu et al., 2021) | 48.06 | 37.95 | 34.92 |
| | | COLT | (Balunovic & Vechev, 2020) | 51.7 | - | 27.5$^{\parallel}$ |
| | | Randomized Smoothing | (Salman et al., 2019a) | 53.0 | - | 24.0$^{\S\parallel}$ |
| | | Randomized Smoothing | (Jeong & Shin, 2020) | 52.3 | - | 25.2$^{\S\parallel}$ |
| | | $\ell_\infty$-distance Net+MLP | (Zhang et al., 2021) | 50.80 | 37.06$^{\ddagger}$ | **35.42** |
| | | $\ell_\infty$-distance Net | (Zhang et al., 2021) | 56.80 | 37.46$^{\ddagger}$ | 33.30 |
| | | $\ell_\infty$-distance Net | This paper | 54.30 | 41.84 | **40.06** |
| | 16/255 | IBP* | (Gowal et al., 2018) | 31.03 | 23.34 | 21.88 |
| | | CROWN-IBP | (Zhang et al., 2020b) | 33.94 | 24.77 | 23.20 |
| | | IBP | (Shi et al., 2021) | 36.65 | - | **24.48** |
| | | $\ell_\infty$-distance Net$^{\dagger}$ | (Zhang et al., 2021) | 55.05 | 26.02$^{\ddagger}$ | 19.28 |
| | | $\ell_\infty$-distance Net | This paper | 48.50 | 32.73 | **29.04** |

* The IBP results are obtained from Zhang et al. (2020b).
$^{\dagger}$ These results are obtained by running the code in the authors' github. See Appendix B for details.
$^{\ddagger}$ The number of PGD steps is chosen as 20 in Zhang et al. (2021).
$^{\S}$ These methods provide probabilistic certified guarantees.
$^{\parallel}$ Calculating certified accuracy for these methods takes several *days* on a single GPU, which is 4 to 6 orders of magnitude slower than other methods.

**Comparing with Zhang et al. (2021).** It can be seen that for all perturbation levels $\epsilon$ and datasets, our proposed training strategy improves the performance of $\ell_\infty$-distance nets. In particular, we boost the certified accuracy on CIFAR-10 from 33.30% to 40.06% under $\epsilon = 8/255$, and from 19.28% to 29.04% under a larger $\epsilon = 16/255$. Note that we use exactly the same architecture as Zhang et al. (2021), and a larger network with better architecture may further improve the results. Another observation from Table 2 is that the improvement of our proposed training strategy gets more prominent with the increase of $\epsilon$. This is consistent with our finding in Section 4.1, in that the optimization is particularly insufficient for large $\epsilon$ using hinge loss, and in this case our proposed objective function can significantly alleviate the problem.

**Comparing with other certification methods.** For most settings in Table 2, our results establish new state-of-the-arts over previous baselines, despite we use the margin-based certification which is *much simpler*. The gap is most noticeable for $\epsilon = 8/255$ on CIFAR-10, where we surpass recent relaxation-based approaches by more than 5 points (Shi et al., 2021; Lyu et al., 2021). It can also be observed that $\ell_\infty$-distance net is most suitable for the case when $\ell_\infty$ perturbation is relatively large. This is not surprising since Lipschitz property is well exhibited in this case. If $\epsilon$ is vanishingly small (e.g. 2/255), the advantage of the Lipschitz property will not be well-exploited and $\ell_\infty$-distance net will face more optimization and generalization problems compared with conventional networks.

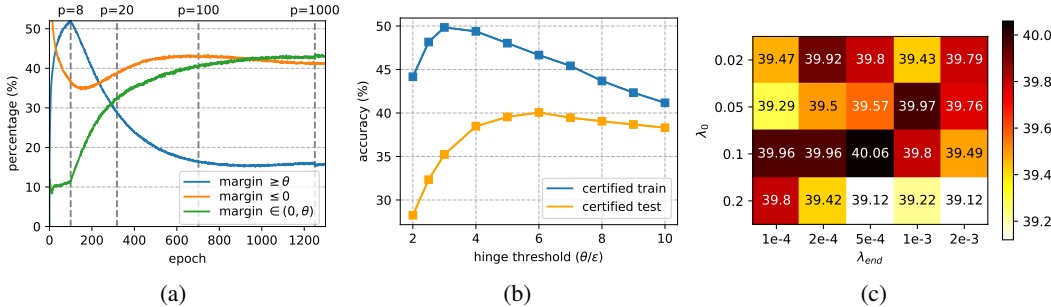

Figure 2: Experiments of $\ell_\infty$-distance net training on CIFAR-10 dataset ($\epsilon = 8/255$) using the proposed objective function (9). (a) The percentage of training samples with output margin greater than $\theta$ (blue), less than 0 (orange), or between 0 and $\theta$ (green) throughout training. The dashed lines indicate different $p$ values of $\ell_p$-relaxation. (b) The final performance of the trained network using different hinge threshold $\theta$. (c) Heatmap of certified accuracy with different hyper-parameters $\lambda_0$ and $\lambda_{\text{end}}$. Each grid shows the certified accuracy for a pair of hyper-parameters.

## 5.3 INVESTIGATING THE PROPOSED TRAINING STRATEGY

We finally demonstrate by experiments that the proposed training strategy indeed addresses the optimization problem in Section 4.1. We first trace the output margin of the training samples throughout training on CIFAR-10 dataset ($\epsilon = 8/255$), and plot the percentage of samples with a margin greater than $\theta$, less than 0, or between 0 and $\theta$, shown in Figure 2(a). In contrast to Figure 1(a), it can be seen that the loss does not degenerate in the whole training process. We then demonstrate in Figure 2(b) that the accuracy curve regarding different choices of hinge threshold $\theta$ is well-behaved compared with Figure 1(b). In particular, the best $\theta$ that maximizes the certified accuracy on training dataset approaches $2\epsilon$ (while for original hinge loss the value is $6\epsilon$). The peak certified accuracy on training dataset also improves by 10 points (see blue lines in the two figures). Such evidence clearly demonstrates the effectiveness of the proposed training strategy.

**Sensitivity analysis**. We perform sensitivity analysis on CIFAR-10 dataset ($\epsilon = 8/255$) over hyper-parameters including the hinge threshold $\theta$ and the mixing coefficient $\lambda$. Results are shown in Figure 2(b) and 2(c). It can be seen that (i) the certified accuracy is above 39% for a wide range of $\theta$ (between $5\epsilon$ and $8\epsilon$); (ii) among the 20 hyper-parameter combinations of $(\lambda_0, \lambda_{\text{end}})$, all certified accuracy results surpass 39%, and half of the results can achieve a certified accuracy of more than 39.75%. In summary, the performance is not sensitive to the choice of the hyper-parameters $\theta$ and $\lambda$. See Appendix F for more details on other hyper-parameters.

**Ablation studies**. We also conduct ablation experiments for the proposed loss function on CIFAR-10 dataset ($\epsilon = 8/255$). Due to the space limit, we put results in Appendix E. In summary, both the cross-entropy loss and clipped hinge loss are crucial to boost the certified accuracy. Using a decaying mixing coefficient $\lambda$ can further improve the performance and stabilize the training.

## 6 RELATED WORK

In recent years substantial efforts have been taken to obtain robust classifiers. Existing approaches mainly fall into two categories: adversarial training and certified defenses.

**Adversarial training.** Adversarial training methods first leverage attack algorithms to generate adversarial examples of the inputs on the fly, then update the model's parameters using these perturbed inputs together with the original labels (Goodfellow et al., 2014; Kurakin et al., 2016; Madry et al., 2017). In particular, Madry et al. (2017) suggested using Projected Gradient Descent (PGD) as the universal attacker to find a perturbation that maximizes the standard cross-entropy loss, which achieves decent empirical robustness. Some recent works considered other training objectives that combine cross-entropy loss and a carefully designed robust surrogate loss (Zhang et al., 2019a; Ding et al., 2020; Wang et al., 2020), which show similarities to this paper. However, all methods above are evaluated empirically using first-order attacks such as PGD, and there is no formal guarantee whether the learned model is truly robust. This motivates researchers to study a new type of method

called certified defenses, in which the prediction is guaranteed to remain the same under all allowed perturbations, thus provably resists against all potential attacks.

**Relaxation-based certified defenses.** These methods adopt convex relaxation to calculate a convex region containing all possible network outputs for a given input under perturbation (Wong & Kolter, 2018; Wong et al., 2018; Dvijotham et al., 2018; 2020; Raghunathan et al., 2018a;b; Weng et al., 2018; Singh et al., 2018; Mirman et al., 2018; Gehr et al., 2018; Wang et al., 2018; 2021; Gowal et al., 2018; Zhang et al., 2018; 2020b; Xiao et al., 2019; Croce et al., 2019; Balunovic & Vechev, 2020; Lee et al., 2020; Dathathri et al., 2020; Xu et al., 2020; Lyu et al., 2021; Shi et al., 2021). If all points in this region correspond to the correct prediction, then the network is provably robust. However, the relaxation procedure is usually complicated and computationally expensive. Furthermore, Salman et al. (2019b); Mirman et al. (2021) indicated that there might be an inherent barrier to tight relaxation for a large class of convex relaxation approaches. This is also reflected in experiments, where the trained model often suffers from severe accuracy drop even on training data.

**Randomized smoothing for certified robustness**. Randomized smoothing provides another way to calculate a probabilistic certification under $\ell_2$ perturbations (Lecuyer et al., 2019; Li et al., 2019a; Cohen et al., 2019; Salman et al., 2019a; Zhai et al., 2020; Jeong & Shin, 2020; Zhang et al., 2020a; Yang et al., 2022; Horváth et al., 2022). For any classifier, if a Gaussian random noise is added to the input, the resulting "smoothed" classifier then possesses a certified guarantee under $\ell_2$ perturbations. Randomized smoothing has been scaled up to ImageNet and achieves state-of-the-art certified accuracy for $\ell_2$ perturbations. However, recent studies imply that it cannot achieve nontrivial certified accuracy for $\ell_p$ perturbations under $\epsilon = \Omega(d^{1/p-1/2})$ when $p > 2$ which depends on the input dimension $d$ (Yang et al., 2020a; Blum et al., 2020; Kumar et al., 2020; Wu et al., 2021). Therefore it is not suitable for $\ell_\infty$ perturbation scenario if $\epsilon$ is not very small.

**Lipschitz networks.** An even simpler way for certified robustness is to use Lipschitz networks, which directly possess margin-based certification. Earlier works in this area mainly regard the Lipschitz property as a kind of regularization and penalize (or constrain) the Lipschitz constant of a conventional ReLU network based on the spectral norms of its weight matrices (Cisse et al., 2017; Yoshida & Miyato, 2017; Gouk et al., 2018; Tsuzuku et al., 2018; Farnia et al., 2019; Qian & Wegman, 2019; Pauli et al., 2021). However, these methods either can not provide certified guarantees or provide a vanishingly small certified radius. Anil et al. (2019) figured out that current Lipschitz networks intrinsically lack expressivity to some simple Lipschitz functions, and designed the first Lipschitz-universal approximator called GroupSort network. Li et al. (2019b); Trockman & Kolter (2021); Singla & Feizi (2021) studied Lipschitz networks for convolutional architectures. Recent studies (Leino et al., 2021; Singla et al., 2022) achieved the state-of-the-art certified robustness using GroupSort network under $\ell_2$ perturbations. However, none of these approaches above can provide good certified results for $\ell_\infty$ robustness. The most relevant work to this paper is Zhang et al. (2021), in which the author designed a novel Lipschitz network with respect to $\ell_\infty$-norm. We show such architecture can establish new state-of-the-art results in the $\ell_\infty$ perturbation scenario.

## 7 CONCLUSION

In this paper, we demonstrate that a simple $\ell_\infty$-distance net suffices for good certified robustness from both theoretical and experimental perspectives. Theoretically, we prove the strong expressive power of $\ell_\infty$-distance nets in robust classification. Combining with Mirman et al. (2021), this result may indicate that $\ell_\infty$-distance nets have inherent advantages over conventional networks for certified robustness. Experimentally, despite simplicity, our approach yields a large gain over previous (possibly more complicated) certification approaches and the trained models establish new state-of-the-art certified robustness.

Despite these promising results, there are still many aspects that remain unexplored. Firstly, in the case when $\epsilon$ is very small, $\ell_\infty$-distance nets may have a lot of room for improvement. Secondly, it is important to design better architectures suitable for image classification tasks than the simple fully connected network used in this paper. Finally, it might be interesting to design better optimization algorithms for $\ell_\infty$-distance nets to further handle the model's non-smoothness and gradient sparsity. We hope this work can make promising the study of $\ell_\infty$-distance nets, and more generally, the global Lipschitz architectures for certified robustness.

ACKNOWLEDGEMENT

This work was supported by National Key R&D Program of China (2018YFB1402600), BJNSF (L172037). Project 2020BD006 supported by PKUBaidu Fund.

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

## A  PROOF OF THEOREM 3.2 AND BEYOND

**Theorem A.1.** *Let $\mathcal{D}$ be a dataset with $n$ elements satisfying the $r$-separation condition with respect to $\ell_\infty$-norm. Then there exists a two-layer $\ell_\infty$-distance net with hidden size $n$, such that the certified $\ell_\infty$ robust accuracy is 100% on $\mathcal{D}$ under perturbation $\epsilon = r$.*

*Proof.* Consider a two layer $\ell_\infty$-distance net $\boldsymbol{g}$ defined in Equation (2). Let its parameters be assigned by

$$
\begin{aligned}
\boldsymbol{w}^{(1,i)} &= \boldsymbol{x}_i, b_i^{(1)} = 0 && \text{for } i \in [n] \\
w_i^{(2,j)} &= C \cdot \mathbb{I}(y_i = j), b_j^{(2)} = -C && \text{for } i \in [n], j \in [K]
\end{aligned}
\tag{10}
$$

where $C = 4 \max_{i \in [n]} \|\boldsymbol{x}_i\|_\infty$ is a constant, and $\mathbb{I}(\cdot)$ is the indicator function. The chosen of $C$ is large enough so that the following holds:

$$
\|\boldsymbol{x}_i - \boldsymbol{x}_j\| \le C/2 \qquad \forall (\boldsymbol{x}_i, y_i), (\boldsymbol{x}_j, y_j) \in \mathcal{D}
\tag{11}
$$

For the above assignment, the first layer simply calculates the $\ell_\infty$-distance between $\boldsymbol{x}$ and each sample in dataset $\mathcal{D}$. We now derive the output of the second layer. We have

$$
\begin{aligned}
[\boldsymbol{g}(\boldsymbol{x})]_j = x_j^{(2)} &= \|\boldsymbol{x}^{(1)} - \boldsymbol{w}^{(2,j)}\|_\infty + b^{(2,j)} \\
&= b^{(2,j)} + \max_{i \in [n]} |x_i^{(1)} - w_i^{(2,j)}| \\
&= -C + \max \left\{ \max_{i \in [n], y_i = j} |x_i^{(1)} - C|, \max_{i \in [n], y_i \ne j} |x_i^{(1)}| \right\} \\
&= -C + \max \left\{ \max_{i \in [n], y_i = j} |\|\boldsymbol{x} - \boldsymbol{x}_i\|_\infty - C|, \max_{i \in [n], y_i \ne j} \|\boldsymbol{x} - \boldsymbol{x}_i\|_\infty \right\} \tag{12} \\
&= -C + \max_{i \in [n], y_i = j} (C - \|\boldsymbol{x} - \boldsymbol{x}_i\|_\infty) \tag{13} \\
&= -\min_{i \in [n], y_i = j} \|\boldsymbol{x} - \boldsymbol{x}_i\|_\infty. \tag{14}
\end{aligned}
$$

Here a core step is Equation (13) which follows by using Inequality (11) when the image $\boldsymbol{x}$ is in dataset $\mathcal{D}$.

From Equation (14) the network $\boldsymbol{g}$ can represent a nearest neighbor classifier, in that it outputs the negative of the nearest neighbor distance between input $\boldsymbol{x}$ and the samples of each class. Therefore, given data $\boldsymbol{x} = \boldsymbol{x}_i$ in dataset $\mathcal{D}$, the output $[\boldsymbol{g}(\boldsymbol{x})]_j$ is either 0 or less than $-2r$ depending on whether $j = y_i$, due to the $r$-separation condition. Therefore the output margin is at least $2r$. In other words, $\boldsymbol{g}$ achieves 100% certified robust accuracy on $\mathcal{D}$. □

We now give a general result which shows that any $L$ layer ($L \ge 2$) $\ell_\infty$-distance net with hidden size $O(n/L + K + d)$ can achieve perfect certified robustness. In this general setting, the total number of neurons in the network is thus $O(n + KL + dL)$ which is still close to real practice.

**Theorem A.2.** *Let $\mathcal{D}$ be a dataset with $n$ elements satisfying the $r$-separation condition with respect to $\ell_\infty$-norm. Then there exists an $L$-layer $\ell_\infty$-distance net with hidden size no more than $\lceil \frac{n}{L-1} \rceil + K + 2d$ where $d$ is the input dimension, such that the certified $\ell_\infty$ robust accuracy is 100% on $\mathcal{D}$ under perturbation $\epsilon = r$.*

*Proof.* The basic idea is to rearrange the computation process of the two-layer network in the above proof by order so as to satisfy the width constraint. To formulate the proof below, we first define some notations. Define $K$ prefix arrays $h_j (j \in [K])$ as follows:

$$
h_{j,k} = -\min_{i \in [k], y_i = j} \|\boldsymbol{x} - \boldsymbol{x}_i\|_\infty.
\tag{15}
$$

Note that we want the network output to be the negative of the nearest neighbor distance of all samples in a class $j$, i.e. $-\min_{i \in [n], y_i = j} \|\boldsymbol{x} - \boldsymbol{x}_i\|_\infty$, which corresponds to $h_{j,n}$. For any hidden layer $\boldsymbol{x}^{(l)}$ ($l \in [L-1]$), we separate it into four sets: $\mathcal{I}^{(l)} = \{x_i^{(l)} : i \in [d]\}, \widetilde{\mathcal{I}}^{(l)} = \{x_i^{(l)} : d < i \le$

$2d\}$, $\mathcal{O}^{(l)} = \{x_i^{(l)} : 2d < i \le 2d + K\}$ and $\mathcal{S}^{(l)}$ containing the rest $\lceil \frac{n}{L-1} \rceil$ neurons. We also denote $\mathcal{O}^{(L)} = \{x_i^{(L)} : i \in [K]\}$ for ease of presentation.

We first make a construction in which the neurons of $\mathcal{I}^{(l)}$ in each layer exactly represent the input $\boldsymbol{x}$, e.g. $x_i^{(l)} = x_i$ ($1 \le i \le d$). This is feasible since an $\ell_\infty$-distance neuron can represent the operation that fetches an element of the neuron input on a bounded domain, e.g. the following operation

$$u(\boldsymbol{z}, \{\boldsymbol{w}, b\}) = \|\boldsymbol{z} - \boldsymbol{w}\|_\infty + b = z_j \quad \forall \boldsymbol{z} \in \mathbb{K} \tag{16}$$

if we assign $w_j = -C, b = -C, w_k = 0 (k \ne j)$ where $C$ is larger than twice the diameter of domain $\mathbb{K}$. In this way, all hidden neurons in the network can have access to the network input $\boldsymbol{x}$. We similarly let the neurons of $\widetilde{\mathcal{I}}^{(l)}$ represent the input $\boldsymbol{x}$ again, e.g. $x_{i+d}^{(l)} = x_i$ ($d < i \le 2d$).

Next, we aim at designing the following computation pattern for $\mathcal{O}^{(l)}$:

$$\mathcal{O}^{(l)} = \{h_{j, \lceil \frac{n(l-1)}{L-1} \rceil} : j \in K\} \quad \text{e.g.} \quad x_{2d+j}^{(l)} = h_{j, \lceil \frac{n(l-1)}{L-1} \rceil}. \tag{17}$$

In this way $\mathcal{O}^{(L)} = \{h_{j,n} : j \in [K]\}$, and the network exactly represents a nearest neighbor classifier which is desired. To represent $\mathcal{O}^{(l+1)}$ in (17), we use the following recursive relation

$$x_{2d+j}^{(l+1)} = h_{j, \lceil \frac{nl}{L-1} \rceil} = \max \left\{ h_{j, \lceil \frac{n(l-1)}{L-1} \rceil}, \max_{\lceil \frac{n(l-1)}{L-1} \rceil < i \le \lceil \frac{nl}{L-1} \rceil, y_i = j} -\|\boldsymbol{x} - \boldsymbol{x}_i\|_\infty \right\}. \tag{18}$$

Note that $h_{j, \lceil \frac{n(l-1)}{L-1} \rceil} = x_{2d+j}^{(l)}$ is already calculated in $\mathcal{O}^{(l)}$ in the previous layer. The left thing is to calculate $\|\boldsymbol{x} - \boldsymbol{x}_i\|_\infty$ for all $i \in \left\{ \lceil \frac{n(l-1)}{L-1} \rceil + 1, \cdots, \lceil \frac{nl}{L-1} \rceil \right\}$, which can be done by the neurons of the set $\mathcal{S}^{(l)}$ in the previous layer (which will be proven later). Assume $\mathcal{S}^{(l)}$ represents

$$\mathcal{S}^{(l)} = \left\{ x_{2d+K+i}^{(l)} : i \in \left[ \lceil \frac{n}{L-1} \rceil \right] \right\}, \quad x_{2d+K+i}^{(l)} = \left\| \boldsymbol{x} - \boldsymbol{x}_{\lceil \frac{n(l-1)}{L-1} \rceil + i} \right\|_\infty, \tag{19}$$

then the neuron $x_{2d+j}^{(l+1)}$ merges the information of neuron $x_{2d+j}^{(l)}$ and part of neurons $x_{2d+K+i}^{(l)}$ in $\mathcal{S}^{(l)}$ depending on whether $y_i = j$, using the construction similar to (10). In detail,

$$x_{2d+j}^{(l+1)} = \|\boldsymbol{x}^{(l)} - \boldsymbol{w}^{(l+1,2d+j)}\|_\infty + b_{2d+j}^{(l+1)} = h_{j, \lceil \frac{nl}{L-1} \rceil} \tag{20}$$

holds by assigning

$$
\begin{aligned}
w_k^{(l+1,2d+j)} &= -C \cdot \mathbb{I}(k = 2d + j) &&\text{for } k \in [2d + K] \\
w_{2d+K+i}^{(l+1,2d+j)} &= C \cdot \mathbb{I}(y_{\lceil \frac{n(l-1)}{L-1} \rceil + i} = j) &&\text{for } i \in \left[ \lceil \frac{n}{L-1} \rceil \right] \\
b_{2d+j}^{(l+1)} &= -C
\end{aligned}
$$

where $C$ is a sufficiently large constant.

Now it remains to represent $\mathcal{S}^{(l)}$ in (19). We first consider the simplest case when $l = 1$. In this case we can directly calculate $x_{2d+K+i}^{(1)} = \|\boldsymbol{x} - \boldsymbol{x}_i\|_\infty$ by assigning proper weights and zero bias. Now assume $l \ge 2$. In this case, we cannot calculate the $\ell_\infty$-distance directly since the previous layer has irrelavant neurons, e.g. the neurons in sets $\mathcal{O}^{(l-1)}$ and $\mathcal{S}^{(l-1)}$. We want to only use the sets of neurons $\mathcal{I}^{(l-1)}$ and $\widetilde{\mathcal{I}}^{(l-1)}$ in the previous layer.

Suppose the objective is to represent $\|\boldsymbol{x} - \boldsymbol{x}_i\|_\infty$ for some $i$. Note that

$$\|\boldsymbol{x} - \boldsymbol{x}_i\|_\infty = \max_{k \in [d]} \max\{x_k - [\boldsymbol{x}_i]_k, [\boldsymbol{x}_i]_k - x_k\}.$$

We assign the parameters of the $\ell_\infty$-distance neuron $x_j^{(l)} = \|\boldsymbol{x}^{(l-1)} - \boldsymbol{w}^{(l,j)}\|_\infty + b_j^{(l)}$ for some $j$ as follows:

$$
\begin{aligned}
w_k^{(l,j)} &= [\boldsymbol{x}_i]_k - C &&\text{for } k \in [d] \\
w_{d+k}^{(l,j)} &= [\boldsymbol{x}_i]_k + C &&\text{for } k \in [d] \\
w_{2d+k}^{(l,j)} &= 0 &&\text{for } k \in \left[ K + \lceil \frac{n}{L-1} \rceil \right] \\
b_j^{(l)} &= -C
\end{aligned}
$$

where $C$ is a sufficiently large constant. In this way

$$
x_j^{(l)} = \|\boldsymbol{x}^{(l-1)} - \boldsymbol{w}^{(l,j)}\|_\infty + b_j^{(l)}
$$

$$
= b_j^{(l)} + \max \left\{ \max_{k \in [d]} |x_k^{(l-1)} - w_k^{(l,j)}|, \max_{k \in [d]} |x_{d+k}^{(l-1)} - w_{d+k}^{(l,j)}|, \max_{k \in \left[K + \lceil \frac{n}{L-1} \rceil\right]} |x_{2d+k}^{(l-1)} - w_{2d+k}^{(l,j)}| \right\}
$$

$$
= -C + \max \left\{ \max_{k \in [d]} (x_k^{(l-1)} - [\boldsymbol{x}_i]_k + C), \max_{k \in [d]} (-x_{d+k}^{(l-1)} + [\boldsymbol{x}_i]_k + C) \right\}
$$

$$
= \max \left\{ \max_{k \in [d]} (x_k - [\boldsymbol{x}_i]_k), \max_{k \in [d]} (-x_k + [\boldsymbol{x}_i]_k) \right\}
$$

$$
= \|\boldsymbol{x} - \boldsymbol{x}_i\|_\infty
$$

which is desired. Proof completes. $\square$

## B  ADDITIONAL EXPERIMENTAL DETAILS AND HYPER-PARAMETERS

Our experiments are implemented using the Pytorch framework. We run all experiments in this paper using a single NVIDIA Tesla-V100 GPU. The CUDA version is 11.2.

The learnable scalar in Equation (9) is initialized to be one and trained using a smaller learning rate that is one-fifth of the base learning rate. This is mainly to make training stable as suggested in Zhang et al. (2019b) since the scalar scales the whole network output. The final performance is not sensitive to the scalar learning rate as long as it is set to a small value. For random crop data augmentation, we use padding = 1 for MNIST and padding = 3 for CIFAR-10. The model is initialized using identity-map initialization (see Section 5.3 in Zhang et al. (2021)), and mean-shift batch normalization is used for all intermediate layers. The training procedure is as follows:

- In the first $e_1$ epochs, we set $p = 8$ in $\ell_p$-relaxation and use $\lambda = \lambda_0$ as the mixing coefficient;

- In the next $e_2$ epochs, $p$ exponentially increases from 8 to 1000. Accordingly, $\lambda$ exponentially decreases from $\lambda_0$ to a vanishing small value $\lambda_{\text{end}}$;

- In the final $e_3$ epochs, $p$ is set to infinity and $\lambda$ is set to 0.

All hyper-parameters are provided in Table 3. Most hyper-parameters are directly borrow from Zhang et al. (2021), e.g. hyper-parameters of the optimizer, the batch size, and the value $p$ in $\ell_p$-relaxation. The only searched hyper-parameters are the hinge threshold $\theta$ and the mixing coefficient $\lambda_0, \lambda_{\text{end}}$. These hyper-parameters are obtained using a course grid search.

Table 3: Hyper-parameters used in this paper.

| Dataset | MNIST | | CIFAR-10 | | |
|---|---|---|---|---|---|
| $\epsilon$ | 0.1 | 0.3 | 2/255 | 8/255 | 16/255 |
| Optimizer | Adam($\beta_1 = 0.9, \beta_2 = 0.99, \epsilon = 10^{-10}$) | | | | |
| Learning rate | 0.03 | | | | |
| Batch size | 512 | | | | |
| $p_{\text{start}}$ | 8 | | | | |
| $p_{\text{end}}$ | 1000 | | | | |
| Epochs | $e_1 = 25, e_2 = 375, e_3 = 50$ | | $e_1 = 100, e_2 = 1150, e_3 = 50$ | | |
| Total Epochs | 450 | | 1300 | | |
| Hinge threshold $\theta$ | 0.6 | 0.9 | 20/255 | 48/255 | 80/255 |
| Mixing coefficient $\lambda_0$ | 0.05 | 0.05 | 0.05 | 0.1 | 0.1 |
| Mixing coefficient $\lambda_{\text{end}}$ | $2 \times 10^{-4}$ | $2 \times 10^{-4}$ | $2 \times 10^{-3}$ | $5 \times 10^{-4}$ | $2 \times 10^{-4}$ |

We also run additional experiments using the training strategy in Zhang et al. (2021) for performance comparison when the original paper does not present the corresponding results. This mainly includes the case $\epsilon = 0.1$ on MNIST and $\epsilon = 2/255, \epsilon = 16/255$ on CIFAR-10, as shown in Table 2. We use the same hyper-parameters in Zhang et al. (2021), except for the hinge threshold $\theta$ where we perform a careful grid search. The choice of $\theta$ is listed in Table 4.

Table 4: Best hinge thresholds for different settings using the training strategy in Zhang et al. (2021).

| Dataset | MNIST | | CIFAR-10 | | |
|---|---|---|---|---|---|
| $\epsilon$ | 0.1 | 0.3 | 2/255 | 8/255 | 16/255 |
| Hinge threshold $\theta$ | 0.8 | 0.9 | 32/255 | 80/255 | 128/255 |

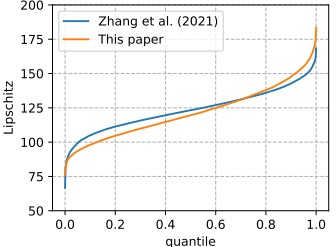

| Loss | Lipschitz | |
|---|---|---|
| | Average | Max |
| Zhang et al. (2021) | 123.5 | 168.4 |
| This paper | 121.5 | 183.4 |

Figure 3: Approximating the Lipschitz constant of $\ell_p$-distance net when $p = 8$, trained using different loss functions on CIFAR-10 dataset. The left figure plots the calculated value of (21) over the test set at each quantile. The right table provides the statistical information.

## C   THE LIPSCHITZ CONSTANT OF $\ell_p$-DISTANCE NET

We have shown in Section 4.1 that an $L$ layer $\ell_p$-distance net with $d$ neurons in each hidden layer is $d^{L/p}$ Lipschitz with respect to $\ell_\infty$-norm. The value becomes quite large if $p$ is small. For example, Zhang et al. (2021) uses a 6-layer $\ell_p$-dist net with $d = 5120$. This gives a Lipschitz constant of approximate 568 when $p = 8$ at the beginning of training.

One may ask whether such *upper bound* of Lipschitz constant (Proposition 4.1) is tight and reflects the true Lipschitz property in practice. To validate the tightness of the bound, we run the following experiments. Consider the $\ell_\infty$-distance net used in Zhang et al. (2021). We train this architecture following the training strategy either in Zhang et al. (2021) or in this paper. After training finishes, we then set $p = 8$ without changing the model parameters. We approximate the Lipschitz constant of the model using Projected Gradient Descent (PGD), which provides a lower bound estimate. In detail, for each image $\boldsymbol{x}$ in the test dataset, we estimate the quantity

$$\frac{1}{\epsilon} \max_{\|\boldsymbol{\delta}\| \leq \epsilon} \|\boldsymbol{g}(\boldsymbol{x} + \boldsymbol{\delta}) - \boldsymbol{g}(\boldsymbol{x})\|_\infty \tag{21}$$

where $\boldsymbol{g}$ is the network and $\epsilon$ is a small constant taken to be 1/255. The expression (21) is clearly a lower bound of the Lipschitz constant (can be seen as the "local Lipschitz constant" near point $\boldsymbol{x}$). It can be further lower bounded by using the PGD solution $\boldsymbol{\delta} = \boldsymbol{\delta}_{\text{PGD}}$. We run PGD for each target label $j$ and optimize

$$\frac{1}{\epsilon} \max_{\|\boldsymbol{\delta}\| \leq \epsilon} |[\boldsymbol{g}(\boldsymbol{x} + \boldsymbol{\delta})]_j - [\boldsymbol{g}(\boldsymbol{x})]_j| \tag{22}$$

using 20 PGD steps with step size $\epsilon/4$.

Results are shown in Figure 3. It can be observed that the "local Lipschitz constant" around *real data points* is indeed far larger than one. The average value exceeds 100 which is close to the theoretical upper bound.

## D   RANDOMIZED SMOOTHING FOR $\ell_\infty$ PERTURBATIONS

Randomized smoothing approaches typically provide probabilistic certified guarantees for $\ell_2$ perturbations. To apply these methods in the $\ell_\infty$ perturbation scenario, most of works convert the result of $\ell_2$ perturbation into $\ell_\infty$ perturbation using norm inequalities (Salman et al., 2019a; Blum et al., 2020). Specifically, to certify the robustness under $\epsilon$-bounded $\ell_\infty$ perturbations, one can certify the robustness under $(\epsilon\sqrt{d})$-bounded $\ell_2$ perturbations to obtain a lower bound estimate where $d$ is the input dimension. On CIFAR-10 dataset, the input dimension $d = 3072$. This corresponds to an $\ell_2$ perturbation radius $\epsilon = 0.4347$ for $\ell_\infty$ perturbation radius $\epsilon = 2/255$, and corresponds to an $\ell_2$ perturbation radius $\epsilon = 1.739$ for $\ell_\infty$ perturbation radius $\epsilon = 8/255$.

For the case $\epsilon = 2/255$, Blum et al. (2020) directly reported a certified accuracy of 62.6% using randomized smoothing which is currently state-of-the-art. For the case $\epsilon = 8/255$, there are no literature results that directly report $\ell_\infty$ robustness, so we use the results of $\ell_2$ robustness from representative papers (Salman et al., 2019a; Jeong & Shin, 2020). Salman et al. (2019a) reported a certified accuracy of 24% and a clean accuracy of 53% under $\ell_2$ perturbation $\epsilon = 1.75$ (Table 17 in their paper). Jeong & Shin (2020) reported a certified accuracy of 25.2% and a clean accuracy of 52.3% under $\ell_2$ perturbation $\epsilon = 1.75$ (Table 1 in their paper, $\sigma = 0.5$). For the case $\epsilon = 16/255$, all randomized smoothing methods fail and only achieve a trivial certified accuracy of 10%.

## E  ABLATION STUDIES

In this section we conduct ablation experiments to the proposed loss. Let a training sample be $(\boldsymbol{x}, y)$ where $y$ is the label of $\boldsymbol{x}$, and denote $\boldsymbol{g}(\boldsymbol{x})$ as the output of an $\ell_\infty$-distance net for input $\boldsymbol{x}$. Let $\ell_{\text{hinge}}(\boldsymbol{z}, y) = \max\{\max_{i \neq y} z_i - z_y + 1, 0\}$ and $\ell_{\text{CE}}(\boldsymbol{z}, y) = \log(\sum_i \exp(z_i)) - z_y$ represent the hinge loss and cross-entropy loss, respectively. We would like to justify that (i) Cross-entropy loss can alleviate the optimization issue in $\ell_p$-relaxation (which is a better substitute over hinge loss), but the threshold of hinge loss is also crucial as it explicitly optimizes the certified accuracy; (ii) Combining cross-entropy loss and clipped hinge loss leads to a much better performance; (ii) Using a decaying mixing coefficient $\lambda$ can further boost the performance and stabilize the training.

We consider the following objective functions:

(1) The baseline hinge loss: $\ell_{\text{hinge}}(\boldsymbol{g}(\boldsymbol{x})/\theta, y)$ with hinge threshold $\theta$. This loss is used in Zhang et al. (2021).

(2) The cross-entropy loss: $\ell_{\text{CE}}(s \cdot \boldsymbol{g}(\boldsymbol{x}), y)$ where $s$ is a scalar (temperature). Note that the information of the allowed perturbation radius $\epsilon$ is not encoded in the loss, and the loss only coarsely enlarges the output margin (see Section 4.2). Therefore it may not achieve desired certified robustness.

(3) A variant of cross-entropy loss with threshold: $\ell_{\text{CE}}(s \cdot \boldsymbol{g}(\boldsymbol{x} - \theta\boldsymbol{1}_y), y)$ where $s$ is a scalar (temperature), $\theta$ is the threshold hyper-parameter and $\boldsymbol{1}_y$ is the one-hot vector with the $y$th element being one. Intuitively speaking, we subtract the $y$th output logit by $\theta$ before taking cross-entropy loss. Compared to the above loss (2), now the information $\epsilon$ is encoded in the threshold hyper-parameter $\theta$. We point out that this loss can be seen as a smooth approximation of the hinge loss.

(4) The combination of cross-entropy loss and clipped hinge loss: $\lambda\ell_{\text{CE}}(s \cdot \boldsymbol{g}(\boldsymbol{x}), y) + \min(\ell_{\text{hinge}}(\boldsymbol{g}(\boldsymbol{x})/\theta, y), 1)$ with a fixed mixing coefficient $\lambda$.

(5) The combination of cross-entropy loss and clipped hinge loss: $\lambda\ell_{\text{CE}}(s \cdot \boldsymbol{g}(\boldsymbol{x}), y) + \min(\ell_{\text{hinge}}(\boldsymbol{g}(\boldsymbol{x})/\theta, y), 1)$ with a decaying $\lambda$. The loss is used in this paper.

We keep the training procedure the same for the different objective functions above. The hyper-parameters such as $\theta$ and $\lambda$ are independently tuned for each objective function to achieve the best certified accuracy. The scalar $s$ is a learnable parameter in each loss except for objective function (2) where we tune the value of $s$. For other hyper-parameters, we use the values in Table 3. We independently run 5 experiments for each setting and the median of the performance is reported. Results are listed in Table 5, and the bracket in Table 5(b) shows the standard deviation over 5 runs.

Table 5: Performance of ablation studies ($\epsilon = 8/255$ on CIFAR-10).

(a) Performance of different objective functions with best hyper-parameters.

| Loss | Clean | Certified | Hyper-parameters |
|------|-------|-----------|------------------|
| (1) | 56.80 | 33.30 | $\theta = 80/255$ |
| (2) | 55.58 | 33.23 | $s = 1.0$ |
| (3) | 53.37 | 34.91 | $\theta = 32/255$ |
| (4) | 53.51 | 39.24 | $\theta = 48/255, \lambda = 0.02$ |
| (5) | 54.30 | **40.06** | $\theta = 48/255, \lambda = 0.1 \to 0$ |

(b) Performance using objective function (4) with different mixing coefficient $\lambda$.

| $\lambda$ | Clean | Certified |
|-----------|-------|-----------|
| 0.1 | 58.99(±0.35) | 37.67(±0.25) |
| 0.05 | 56.50(±0.25) | 38.82(±0.14) |
| 0.02 | 53.51(±0.40) | **39.24(±0.37)** |
| 0.01 | 50.48(±1.83) | 38.05(±1.31) |
| 0.005 | 47.51(±5.03) | 37.03(±2.64) |
| 0 | 10.0 | 10.0 |

We can draw the following conclusions from Table 5:

- Hinge loss and cross-entropy loss are complementary. Cross-entropy is better in the early training phase when the Lipschitz constant is large, while hinge loss is better for certified robustness when the model is almost 1-Lipschitz in the later training phase. This can be seen from the results of objective functions (1-3) in Table 5(a), where (3) incorporates cross-entropy loss and the threshold in hinge loss, and outperforms both (1) and (2) by a comparable margin.

- Combining cross-entropy loss and clipped hinge loss leads to much better performance. This can be seen from the result of the objective function (4), which significantly outperforms (1-3). However, this loss is very sensitive to the hyper-parameter $\lambda$ as is demonstrated in Table 5(b). If $\lambda$ is too large, the certified accuracy gets worse. If $\lambda$ is too small, the training becomes unstable and the clean accuracy drops significantly. In the extreme case when $\lambda = 0$, the loss (4) reduces to the clipped hinge loss and the optimization fails because clipped hinge loss does not optimize for wrongly-classified samples.

- Using a decaying mixing coefficient $\lambda$ can further boost the performance and stabilize the training. In contrast to the loss (4), we will show in Appendix F that the proposed objective function (5) in this paper is not sensitive to hyper-parameter $\lambda$.

## F   SENSITIVITY ANALYSIS

In this section we provide sensitive analysis of the proposed objective function (9) with respect to hyper-parameters. We consider the setting $\epsilon = 8/255$ on CIFAR-10 dataset. For each choice of hyper-parameters, we independently run 3 experiments and the median of the certified accuracy is reported.

**The hinge threshold $\theta$.** The results are already plotted in Figure 2(b). We list the concrete numbers below.

Table 6: Sensitive Analysis over hyper-parameter $\theta$.

| $\theta$ | $3\epsilon$ | $4\epsilon$ | $5\epsilon$ | $6\epsilon$ | $7\epsilon$ | $8\epsilon$ | $9\epsilon$ | $10\epsilon$ |
|---|---|---|---|---|---|---|---|---|
| Certified | 35.23 | 38.47 | 39.55 | 40.06 | 39.46 | 39.05 | 38.68 | 38.31 |

**The mixing coefficients $\lambda_0$ and $\lambda_{\text{end}}$.** The results are already shown in Figure 2(c).

**The number of epochs**. Our best result reported in Table 2 is trained for 1300 epochs, which is longer than Zhang et al. (2021). We also consider using the same training budget by setting $e_1 = 100, e_2 = 650, e_3 = 50$ in Table 3. This yields a total of 800 training epochs. In this way we can achieve 54.52 clean accuracy and 39.61 certified accuracy.

**Adam hyper-parameters**. While we use the same Adam hyper-parameters as Zhang et al. (2021), note that the values are different from the default numbers in Pytorch (e.g. $\beta_2 = 0.999$ and $\epsilon = 10^{-8}$). Instead, we use $\beta_2 = 0.99$ and $\epsilon = 10^{-10}$ in all experiments.

For $\epsilon$, we find the value is essential to be small, because for the $\ell_p$-distance function, the gradients of most of the elements are close to zero when $p$ is large. For Adam optimizer (Kingma & Ba, 2015), the update is written as

$$w_{t+1} = w_t - lr \cdot \frac{m_t}{\sqrt{v_t} + \epsilon}. \tag{23}$$

Then a large $\epsilon$ will dominate the denominator in Adam if $v_t$ is close to zero, which severely weakens the parameters' update and leads to worse performance. We find the value of $\epsilon$ is not sensitive if it is small enough, i.e. we can obtain almost identical performance for $\epsilon = 10^{-10}, 10^{-11}$ or $10^{-12}$.

The use of a smaller $\beta_2$ is mainly because of the $\ell_p$ relaxation. Since $p$ increases exponentially during training, the network function changes through time, therefore the long-time-ago historical gradient information will become meaningless and even do harm to the training. This is why a smaller $\beta_2$ is used, so that the second-order momentum (the term $v_t$) in Adam only depends on recent gradient information. We find it is also OK to choice $\beta_2 = 0.98$ or $0.995$, and the certified accuracy can reach 39.5%. However, the default value of 0.999 is too large, as it corresponds to 10 times more historical information than the value of 0.99.

