# OpenReview forum: "Boosting the Certified Robustness of L-infinity Distance Nets"
_ICLR.cc/2022/Conference — ICLR 2022 Poster_

### Official Review · Reviewer_TteL · 2021-10-20

**Correctness:** 3
**Technical Novelty And Significance:** 3
**Empirical Novelty And Significance:** 3
**Recommendation:** 8
**Confidence:** 4

**Main Review:**

Strength:
1. It is very impressive to achieve 40% certified accuracy for 8/255 attack. The result significantly close the gap between the state-of-the-art of empirical robustness (~53% by TRADES) and that of certified robustness.
2. The paper has a thorough analysis for the training problem in Zhang et al. (2021), and proposes a new method to resolve the issue.
3. The theoretical analysis of expressive power of l_infty distance net is interesting, though it is a little out of the selling point of this paper.

Weakness:
1. The 2/255 experiment in Table 2 does not compare with randomized smoothing. Randomized smoothing is able to achieve 62.6% certified robustness for 2/255 radius (see Table 1 in [1]), which outperforms the proposed method by ~8%.
[1] https://arxiv.org/pdf/2002.03517.pdf

Questions:
1. Is there a particular reason that in Table 2, the 2/255, 8/255 and 16/255 experiments use different l_infty distance nets (they have different clean accuracy)?

2. We know that adversarial training helps in randomized smoothing (see [2]). Does adversarial training helps in improving the certified robustness of l_infty distance net?
[2] https://arxiv.org/pdf/1906.04584.pdf

**Summary Of The Paper:**

This paper is a follow-up paper of Zhang et al. (2021). In Zhang et al. (2021), the authors proposed a new network architecture, l_infty distance net. By construction, the network is 1-Lipschitz w.r.t. l_infty distance. However, the training procedure therein is problematic. This paper resolves the issue by a new loss design of scaled cross entropy loss + clipped hinge loss. Without using MLP on top of the l_infty distance net backbone, the proposed new training method outperforms the original one in Zhang et al. (2021) and improves over the state-of-the-art by more than 5% for 8/255 and other radiuses. Theoretically, the paper shows the expressive power of l_infty distance net for well-separated data.

**Summary Of The Review:**

Overall, I like the paper (in particular, the design of l_infty distance net, which is not the focus of this paper but of Zhang et al. (2021)). The design goes towards solving the robustness issue by new network design, rather than regarding the neural network as a black-box. The experiment result on 8/255 is promising and exciting.

---

> ### Author Response · Authors · 2021-11-16
> **Response to Reviewer TteL**
>
> We sincerely thank the reviewer for the positive feedback and valuable suggestions. Below are our responses to the reviewer's questions.
>
> **Response to Weakness 1**. Thanks for pointing it out. We will add the comparison and discussion in the next version of the paper. Indeed, we observed that $\ell_\infty$-distance nets under-perform randomized smoothing when the perturbation level $\epsilon$ is very small (2/255) but outperform it when $\epsilon$ is relatively large (8/255, 16/255). We think it is interesting to study these phenomena and are investigating the reason behind them.
>
> **Response to Question 1**. Thanks for the question. When we target different perturbation levels, we train $\ell_\infty$-distance nets using different hinge thresholds $\theta$ (see Table 3). For a larger perturbation $\epsilon$, we have to use a larger value of hinge threshold $\theta$ to better guarantee the model's robustness. The different values of $\theta$ therefore lead to different clean accuracy of the trained models.
>
> **Response to Question 2**. Thank you for the valuable suggestion. We have tried to use adversarial training for $\ell_\infty$-distance nets but currently we cannot see a significant improvement. We hypothesize the reason is that the optimization of the $\ell_\infty$-distance net model is challenging. For example, it requires much more PGD iterations to generate adversarial examples for $\ell_\infty$-distance nets. Therefore further incorporating adversarial attacks during training will make the optimization more difficult. Nevertheless, we believe it is a promising research direction if the optimization difficulty can be well-handled, and we will keep investigating this in future work.

---

### Official Review · Reviewer_THnD · 2021-11-02

**Correctness:** 4
**Technical Novelty And Significance:** 3
**Empirical Novelty And Significance:** 4
**Recommendation:** 6
**Confidence:** 4

**Main Review:**

The paper is well-written and the improvement on empirical result is impressive.
My main concern is on the experiments.
* The paper proposed multiple modifications on the loss function, including the scale factor on cross-entropy, clipped hinge loss and weight coefficient. The authors should provide comprehensive ablation study on the several modifications.
* I recommend the authors to include sensitive analysis for the hyper-parameter $\lambda$ and hinge threshold $\theta$, which are important if we want to apply this method to different models.
* For reproducibility, since the code is not provided, I recommend the author to provide a detailed instruction on how to reproduce the paper based on https://github.com/zbh2047/L_inf-dist-net.
* After checking the hyper parameters, I found that the authors have tuned the beta_2 and eps of Adam optimizer. Is there any explanation for this tuning?


**Summary Of The Paper:**

The paper proposed a new loss to improve the performance of Linf distance network a new network for Linf robustness and achieved impressive empirical results.

**Summary Of The Review:**

The paper is well-written and the proposed method has an impressive empirical performance. My concern is mainly on the experiments and reproducibility.

---

> ### Author Response · Authors · 2021-11-16
> **Response to Reviewer THnD (Part 1: Ablation Study)**
>
> We thank the reviewer for raising these concerns on experiments. Below are our responses to the reviewer's concerns.
>
> **Ablation study**. Thanks for the suggestion. We performed a complete ablation study of the proposed loss function following your advice. Details are demonstrated below.
>
> We consider five cases: (1) using the baseline hinge loss; (1) using the vanilla cross-entropy loss; (2) using the clipped hinge loss; (3) using the combination of the scaled cross-entropy and clipped hinge loss, with a fixed weight coefficient; (4) using the combination of the scaled cross-entropy and clipped hinge loss, with a decaying weight coefficient. The results are listed below.
>
> | loss | clean acc | certified acc |
> | ---- | --------- | ------------- |
> | (1)  | 56.80     | 33.30         |
> | (2)  | 55.58     | 33.23         |
> | (3)  | 10.00     | 10.00         |
> | (4)  | 53.51     | 39.24         |
> | (5)  | 54.30     | 40.06         |
>
> Firstly, we can observe that using cross-entropy alone cannot achieve better certified robustness. This is obviously true since the information $\epsilon$ (perturbation radius) is not encoded in the cross-entropy term. We therefore make a further investigation of the cross-entropy loss by performing the following experiment, where for an input image with label $y$ we subtract the $y$th output logit by $\theta$ before taking cross-entropy loss (here $\theta$ is a hyper-parameter depending on $\epsilon$). Using such a naive loss we are already able to achieve 34.91% certified accuracy which is noticeably higher than the baseline result 33.30% using hinge loss.
>
> Secondly, combining the scaled cross-entropy loss and clipped hinge loss leads to a much better performance than using the individual component. We perform experiments using the trivial combination of scaled cross-entropy loss and clipped hinge loss with a fixed weight coefficient. Detailed results are listed below. It can be seen that 1) If the weight coefficient is large, the clean accuracy is better but the certified accuracy becomes worse; 2) If the weight coefficient is extremely small, the training is unstable and the clean accuracy drops drastically. For the extreme case when the $\lambda$ is zero, the loss degenerates to the clipped hinge, which fails to optimize well since clipped hinge does not take effect for wrongly classified samples. 3) The best weight coefficient ($\lambda=0.02$) achieves 39.24 certified accuracy. Therefore using a decaying weight coefficient $\lambda$ (decayed from 0.1 to a vanishing value) is better than using a fixed $\lambda$.
>
>
> | wight coefficient $\lambda$ | clean acc  | certified acc |
> | --------------------------- | ---------- | ------------- |
> | 0.1                         | 58.99(±0.35) | 37.67(±0.25)    |
> | 0.05                        | 56.50(±0.25) | 38.82(±0.14)    |
> | 0.02                        | 53.51(±0.40) | 39.24(±0.37)    |
> | 0.01                        | 50.48(±1.83) | 38.05(±1.31)    |
> | 0.005                       | 47.51(±5.03) | 37.03(±2.64)    |
> | 0                           | 10.00      | 10.00         |
>
> In summary, to achieve the best performance, all three components in our loss function are necessary. We will add the ablation study in the next version of this paper.

---

> > ### Comment · Reviewer_THnD · 2021-11-29
> > **Thank you for the response.**
> >
> > Thank you for the detailed response! I appreciate the additional experiments and I have two remaining questions.
> > 1. Why do you choose clipped hinge loss while the baseline hinge loss alone is better than the clipped hinge loss? Do you have experiment result for "scaled cross-entropy + baseline hinge loss"?
> > 2. The reason that I asked about parameters of Adam optimizer is that I found the performance of your work is highly sensitive to beta_2 and eps in my implementation. Do you have an explanation for this?

---

> > > ### Author Response · Authors · 2021-11-29
> > > **Response to remaining questions**
> > >
> > > Thank you for the feedback. We answer each of these questions below.
> > >
> > > **To Question 1**:  We use clipped hinge loss because it is a better choice when used together with the cross-entropy loss, see the 5th paragraph on Page 6. Note that the cross-entropy loss already focuses on learning a model with high (clean) accuracy. Clipped hinge loss then optimizes the certified radius (our goal) for the *correctly classified samples* in the late train phase when $p$ is large, and ignores the wrongly classified ones that may have little chance to be robust since the optimization already becomes difficult ($p$ approaching infinity). We have run cross-entropy + hinge loss in the past, and results show that it is able to outperform vanilla hinge loss or the cross-entropy loss, but much worse than cross-entropy + clipped hinge. We can add the result in the next version of this paper.
> > >
> > > **To Question 2**: Thanks for the question. We observe that a very small $\epsilon$ value ($10^{-10}$) is used in the experiments. We find the value is essential to be small, because for the $\ell_p$-distance function, the gradients of most of the elements are close to zero when $p$ is large. For Adam optimizer, the update is written as $$w_{t+1}=w_t-lr\cdot \frac {m_t} {\sqrt{v_t}+\epsilon}.$$ Then a large $\epsilon$ will dominate the denominator in Adam if $v_t$ is close to zero, which severely weakens the parameters' update and leads to worse performance. We find the value of $\epsilon$ is not sensitive if it is small enough, i.e. we can obtain almost identical performance for $\epsilon=10^{-10}$, $10^{-11}$ or $10^{-12}$. This result is reproducible.
> > >
> > > The use of a smaller $\beta_2$ is mainly because of the $\ell_p$ relaxation. Since $p$ increases exponentially during training, the network function changes through time, therefore the long-time-ago historical gradient information will become meaningless and even do harm to the training. This is why a smaller $\beta_2$ is used, so that the second-order momentum (the term $v_t$) in Adam only depends on recent gradient information.
> > >
> > > When a reasonable $\epsilon$ is used as above, the choice of $\beta_2$ is not sensitive. For example, when choosing $\beta_2=0.98$ or $\beta_2=0.995$, the certified accuracy can reach 39.5% using different seeds. However, the default value of 0.999 is too large, as it corresponds to 10 times more historical information than the value of 0.99.

---

> > > > ### Comment · Reviewer_THnD · 2021-11-29
> > > > **Thank you for your rebuttal.**
> > > >
> > > > Thank you for your response, I am willing to increase my score.

---

> ### Author Response · Authors · 2021-11-16
> **Response to Reviewer THnD (Part 2: Sensitive Analysis, Reproducibility and Adam Hyper-parameters)**
>
> We thank the reviewer for raising these concerns on experiments. Below are our responses to the reviewer's concerns.
>
> **Sensitive analysis for the hyper-parameters**. We have performed experiments to see the sensitivity over hyper-parameters following your advice. For the weight coefficient $\lambda$, the results are listed in the table below. Each grid in the table shows the certified robust accuracy for a pair of hyper-parameters $(\lambda_0,\lambda_{end})$. We independently run 3 experiments for each case and the median of the certified accuracy is reported. It can be seen that among the 20 hyper-parameter combinations, half of the results can achieve a certified accuracy of more than 39.7% (indicated in boldface in the table below). In other words, the performance is not sensitive to the choice of the $\lambda$.
>
> | $\lambda_0$\ $\lambda_{end}$ | 1e-4      | 2e-4      | 5e-4      | 1e-3      | 2e-3      |
> | ---------------------------- | --------- | --------- | --------- | --------- | --------- |
> | 0.02                         | 39.47     | **39.92** | **39.80** | 39.43     | **39.79**|
> | 0.05                         | 39.29     | 39.50     | 39.57     | **39.97** | **39.76** |
> | 0.1                          | **39.96** | **39.96** | **40.06** | **39.80** | 39.49     |
> | 0.2                          | **39.80** | 39.42     | 39.12     | 39.22     | 39.12     |
>
> For the sensitivity of the hinge threshold $\theta$, the results are already shown in Figure 2(b) of the paper. We list concrete numbers below.
>
> | Hinge threshold | certified acc |
> | ----- | -------- |
> |$3\epsilon$  | 35.23 |
> | $4\epsilon$  | 38.47 |
> | $5\epsilon$  | 39.55 |
> | $6\epsilon$  | 40.06 |
> | $7\epsilon$  | 39.51 |
> | $8\epsilon$  | 39.05 |
>
> **Reproducibility**. Our code is indeed based on https://github.com/zbh2047/L_inf-dist-net. We have put the training logs of all the experiments in the supplementary material for the sake of transparency. We are cleaning up the code and will make it publicly available soon. We will upload the code and provide detailed commands as a supplementary material before the discussion period ends.
>
> **Hyper-parameters of Adam optimizer**. We didn't tune the Adam hyper-parameters. We strictly follow the choice of Adam hyper-parameters in Zhang et al. (2021)[1], e.g. $\beta_2=0.99$ and $\epsilon=10^{-10}$ (see Table 6 in [1]).
>
> We hope our response above can address your questions and concerns, and we sincerely hope the reviewer can reevaluate our paper based on our detailed response.
>
>
>
> [1] Bohang Zhang, Tianle Cai, Zhou Lu, Di He, and Liwei Wang. Towards certifying l-infinity robustness using neural networks with l-inf-dist neurons. In ICML, 2021.

---

> ### Author Response · Authors · 2021-11-27
> **Thank you again for the review and hope the reviewer can check out our response**
>
> Dear Reviewer THnD,
>
> Thank you again for your detailed review. As the discussion period is closing soon, we hope the reviewer can take a look at our response and reevaluate our paper based on the detailed response along with the revised paper.
>
> In our response, we provided a complete ablation study, added the sensitivity analysis for hyper-parameters, and uploaded the code with a detailed command. We believe our answer can address your concern about the experiments and reproducibility.
>
> Let us know if you have further questions about our paper and we look forward to hearing from you.
>
> Sincerely,
> Paper 3247 Authors

---

### Official Review · Reviewer_WMpC · 2021-11-02

**Correctness:** 3
**Technical Novelty And Significance:** 3
**Empirical Novelty And Significance:** 2
**Recommendation:** 5
**Confidence:** 3

**Main Review:**

Strengthens:
1. The paper overall is easy to read and follow.
2. The authors study an important problem of improving the robustness of deep network from the perspective of reegurozation.


Weakness:
1. Moderate novelty. This paper follows a similar setting with prior work (Zhang et al. 2021) published in ICML 2021, specially targeted in $l_\infty$ distance net for improving certified accuracy. The proposed method serves as a regularization term for the hinge loss to train a more robust network. Apart from that, this work has moderate novelty which shows a large portion of prior work with the same setting and experiment design.

2. Limited experimental design. To the best of my knowledge, this work shares the same setup with prior work where experiments are not extensive enough. For example, the empirical study is limited to improving the robustness in small datasets such as MNIST. What about other bigger datasets? And also, is this proposed method applicable to other machine learning tasks other than the listed task in the paper?

**Summary Of The Paper:**

The authors study to improve the certified robustness of $l_\infty$ distance nets by introducing a regularization term to train the network. Guided by the fundamental assumption of "existing a two-layer $l_\infty$ net which has 100% robust accuracy", this method shows better performance over other baselines.

**Summary Of The Review:**

This paper is largely based on prior work (Zhang et al. (2021))  which follows the same setting and experimental design. The authors propose to prove the effectiveness of $l_\infty$-distance net where they come up with a regularized hinge loss to learn robust classifiers guided by their proposed Theorem 3.2.

---

> ### Author Response · Authors · 2021-11-16
> **Response to Reviewer WMpC**
>
> We thank the reviewer for raising these concerns and below are our responses to each of these concerns.
>
> **Regarding the novelty of this work**.  We respectfully disagree with the reviewer about the novelty of this paper. We believe the proposed method is novel, and the empirical results are significant.
>
> The main contribution of the work is to investigate and solve optimization issues in Zhang et al. (2021) [4], which is largely under-explored. We first find that the original loss fails to optimize well by providing thorough evidence. We then propose a new training method with a well-motivated modification of the previous training strategy (Section 4), which demonstrates that the certified robust accuracy can be dramatically improved. We agree that our approach is simple (i.e., following most of the training configuration of Zhang et al. (2021) [4] with moderate modification), but such simplicity should be regarded as an advantage rather than a drawback given the superior performance.
>
> **Regarding more datasets and tasks**. To further show the effectiveness of the proposed training methods, during the discussion period we have run experiments on the TinyImageNet dataset with a challenging perturbation $\epsilon=8/255$.  This task is challenging since SOTA robust training methods only achieve 13.56% empirical robust accuracy under AutoAttack [1], and the baseline adversarial training method (Madry et al. 2017) [2] only achieves 9.20% empirical robust accuracy under AutoAttack. We also try the current SOTA method for certified robustness [3,5] using their github repos and commands. We run the code independently for 4 times and pick the best number. Using the CNN7+BN model, Xu et al. (2020)[5] achieve 9.24% clean accuracy and 4.02% certified accuracy using CROWN-IBP with loss fusion. Shi et al. (2021)[3] achieved 10.31% clean accuracy and 4.56% certified accuracy using the CNN7+FullBN model with IBP initialization and regularization.
>
> As a comparison, the $\ell_\infty$-distance net can achieve 11.01% clean accuracy and 5.78% certified accuracy using the method in this paper. Moreover, our model is 3 times smaller than the CNN7+FullBN model in [3] (156M FLOPs vs. 458M FLOPs). We think these results demonstrate that $\ell_\infty$-distance nets trained using our approach can still outperform previous SOTA results on larger datasets.
>
> This paper focuses on certified robustness for classification tasks, and we believe the results on three benchmark datasets MNIST, CIFAR-10 and TinyImageNet suffice to show the effectiveness of $\ell_\infty$-distance nets in the classification setting. As for other tasks, such as language modeling in NLP or object detection in CV, it is not very related to the main purpose of this paper, and we will leave them as future research.
>
> We hope our response above can address your questions and concerns, and we sincerely hope the reviewer can reevaluate the paper based on our response.
>
>
> [1] Jihoon Tack, Sihyun Yu, Jongheon Jeong, Minseon Kim, Sung Ju Hwang and Jinwoo Shin. Consistency Regularization for Adversarial Robustness. In ICML Workshop on Adversarial Machine Learning, 2021.
>
> [2] Aleksander Madry, Aleksandar Makelov, Ludwig Schmidt, Dimitris Tsipras, and Adrian Vladu. Towards deep learning models resistant to adversarial attacks. In ICLR, 2018.
>
> [3] Zhouxing Shi, Yihan Wang, Huan Zhang, Jinfeng Yi, and Cho-Jui Hsieh. Fast certified robust training with short warmup. In NeurIPS, 2021.
>
> [4] Bohang Zhang, Tianle Cai, Zhou Lu, Di He, and Liwei Wang. Towards certifying l-infinity robustness using neural networks with l-inf-dist neurons. In ICML, 2021.
>
> [5] Kaidi Xu, Zhouxing Shi, Huan Zhang, Yihan Wang, Kai-Wei Chang, Minlie Huang, Bhavya Kailkhura, Xue Lin, and Cho-Jui Hsieh. Automatic perturbation analysis for scalable certified robustness and beyond. In NeurIPS, 2020.

---

> ### Author Response · Authors · 2021-11-27
> **Thank you again for the review and hope the reviewer can check out our response**
>
> Dear Reviewer WMpC,
>
> Thank you again for your detailed review. As the discussion period is closing soon, we hope the reviewer can take a look at our response and reevaluate our paper based on the detailed response along with the revised paper.
>
> In our response, we clarified our contribution to this work and added detailed results on the large TinyImageNet dataset. In the revised paper we also performed an additional ablation study. We believe the experiments in this paper are extensive enough now (including multiple datasets, multiple perturbation radii, ablation study, and sensitivity analysis).
>
> Let us know if you have further questions about our paper and we look forward to hearing from you.
>
> Sincerely,
> Paper 3247 Authors

---

### Official Review · Reviewer_d51P · 2021-11-03

**Correctness:** 3
**Technical Novelty And Significance:** 3
**Empirical Novelty And Significance:** 4
**Recommendation:** 6
**Confidence:** 4

**Main Review:**

Major comments:
- The main focus of the paper is on the training of $\ell_\infty$ net. The authors emphasize that the original training is hindered by the  $\ell_p$ relaxation, and they use clipped hinge loss to relieve the issue. The authors also implemented scaled cross-entropy loss. From my point of view, the authors do not explain intuitions very well. Why clipped hinge loss can maintain the Lipschitz property of $\ell_\infty$ even when $\ell_p$ relaxation is used? How should we decompose the effect of scaled cross-entropy and clipped hinge loss?
- The empirical results of this paper seem pretty decent to me. The certified accuracy is able to beat IBP. I wonder how it compares to randomized smoothing?

Minor comments:
- When you search on the hyper-parameters, is any cross-validation performed?
- How do you explain the generalization gap between the certified train and certified test? It seems larger than the previous training scheme.
- How large is $\theta$ in Figure 1(a) and 2(a)?

**Summary Of The Paper:**

This paper proposed a simple modification of $\ell_\infty$ net training, which boosts the accuracy for certified robustness under $\ell_\infty$ attack. It provides a trainable scale on the output of the network and uses a clipped hinge loss. The paper also proves the expressive ability of $\ell_\infty$ nets for classification problems.

**Summary Of The Review:**

My current assessment of this paper is slightly below the acceptance threshold because I think the intuitions are not well explained and there are questions that I think the authors should explain to readers.

---

> ### Author Response · Authors · 2021-11-16
> **Response to Reviewer d51P**
>
> We thank the reviewer for the detailed comments and answer each of these comments below.
>
> **Regarding the loss design**. We would like to clarify that the reviewer may have some misunderstandings about our loss design. It is the cross-entropy loss (not the clipped hinge loss) that aims to ''relieve'' the $\ell_p$ relaxation issue (see the first two paragraphs in Section 4.2). We would like to further explain our intuitions below.
>
> In detail, we observe that $\ell_p$ relaxation leads to a large Lipschitz constant at the early training stage when $p$ is small. In this case, the optimization using hinge loss is insufficient: even if the margin of a data point passes the hinge threshold (thus achieving zero loss), the data point can still lie near to classification boundary (see the last paragraph on Page 5). Based on this observation, to make the optimization sufficient, the output margin should be much larger if the Lipschitz constant is large. This is why we propose to use cross-entropy loss to make the optimization sufficient at the early stage since cross-entropy will always push the output margin towards being as large as possible.
>
> The clipped hinge loss is designed to achieve robustness when $p$ is already large (i.e., when the Lipschitz constant approaches one). See the second paragraph in Section 4.2. This term is exactly the surrogate of 0-1 certified robust error. It focuses on improving the certified accuracy by optimizing the output margin for correctly classified samples.
>
> We believe such misunderstanding is partially due to the presentation of section 4.2, which is not clear enough.  We will re-organize the related discussions in the next version of this paper. We also hope our explanation can help the reviewer better evaluate our work.
>
> **Comparing to Randomized smoothing**. Randomized smoothing approaches usually cannot produce a good certification guarantee for the commonly used $\ell_\infty$ perturbation radii, both empirically and theoretically, as demonstrated in [2]. For example,  for commonly used $\epsilon=8/255$ on CIFAR-10 dataset, Salmon et al. (2019)[1] reported a certified accuracy of approximately 24% (Table 2 in their paper, transformed from $\ell_2$-norm results); Jeong and Shin (2020)[3] reported a certified accuracy of approximately 25.2%  (Table 1 in their paper). Both of them are lower than the baseline approaches used in our paper. We can add these baselines in the next version of this paper if needed.
>
> **Regarding hyper-parameter selection**. We followed Zhang et al. (2021) [4] for most hyper-parameters. The only searched hyper-parameters are the hinge threshold $\theta$ and the mixing coefficient $\lambda$. These hyper-parameters are obtained using a *very coarse* grid search (see Table 3) and are picked using ten-fold cross-validation. We find that the certified accuracy is not sensitive to hyper-parameter $\lambda$.
>
> **Regarding the generalization gap**. With our training approach, the generalization gap does not change much or even decreases compared to the previous training scheme when using a typical hinge threshold $\theta$, for example, $\theta=6\epsilon$ in this paper. We list the exact number below (obtained from Figure 1(b) and Figure 2(b)):
>
> | $\theta$    | previous generalization gap | this paper       |
> | -------- | --------------- | ---------------- |
> | $5\epsilon$ | 39.00-30.21=8.79            | 48.03-39.55=8.48 |
> | $6\epsilon$ | 39.92-30.79=9.13            | 46.66-40.06=6.40 |
> | $7\epsilon$ | 39.88-31.76=8.12            | 45.42-39.46=5.96 |
> | $8\epsilon$ | 39.39-32.70=6.69            | 43.69-39.05=4.64 |
>
> **Regarding the hinge threshold in Figure 1 and Figure 2**. In Figure 1(a) the hinge threshold is chosen as $\theta=10\epsilon=80/255$ according to Zhang et al. (2021) [4]. In Figure 2(a) the hinge threshold is  $\theta=6\epsilon=48/255$. We have explained why a large hinge threshold has to be used in Section 4.1, and why the training approach this paper proposed enables us to use a smaller hinge threshold in Section 4.2.
>
> We hope our explanation above can address your questions and concerns, and we sincerely hope the reviewer can reevaluate the paper based on our response.
>
> [1] Hadi Salman, Greg Yang, Jerry Li, Pengchuan Zhang, Huan Zhang, Ilya Razenshteyn, and Sebastien Bubeck. Provably robust deep learning via adversarially trained smoothed classifiers. In ICML, 2019.
>
> [2] Avrim Blum, Travis Dick, Naren Manoj, and Hongyang Zhang. Random Smoothing Might be Unable to Certify $\ell_\infty$ Robustness for High-Dimensional Images. arXiv:2002.03517, 2017.
>
> [3] Jongheon Jeong and Jinwoo Shin. Consistency regularization for certified robustness of smoothed classifiers. In NeurIPS, 2020.
>
> [4] Bohang Zhang, Tianle Cai, Zhou Lu, Di He, and Liwei Wang. Towards certifying l-infinity robustness using neural networks with l-inf-dist neurons. In ICML, 2021.

---

> > ### Comment · Reviewer_d51P · 2021-11-29
> > **Response to updated draft**
> >
> > Thank the authors for their thorough response to reviewer comments and updates to the paper. The authors clarified most of my concerns, especially on the intuition of loss design. I've updated my score to reflect this.

---

> ### Author Response · Authors · 2021-11-27
> **Thank you again for the review and hope the reviewer can check out our response**
>
> Dear Reviewer d51P,
>
> Thank you again for your detailed review. As the discussion period is closing soon, we hope the reviewer can take a look at our response and reevaluate our paper based on the detailed response along with the revised paper.
>
> In our response, we clarified our loss design and added comparisons with randomized smoothing. We also answered each of your minor comments. In the revised paper we performed an additional sensitivity analysis over hyper-parameters.
>
> Let us know if you have further questions about our paper and we look forward to hearing from you.
>
> Sincerely,
> Paper 3247 Authors

---

### Author Response · Authors · 2021-11-22
**We have updated the paper and hope to hear from the reviewers**

We would like to thank all reviewers again for the insightful comments and valuable feedback. We follow the comments and suggestions raised by reviewers to update the paper. Below, we briefly outline the major updates of the revised submission:

**Paper updates**:
- [Section 4.2] We revise this section by adding more intuition of our loss design and providing a detailed explanation about the role of scaled cross-entropy loss and clipped hinge loss (reviewer d51P). We hope the contribution and novelty of this paper can be better clarified after the revision (reviewer WMpC).
- [Table 2 and Appendix D] We add the comparison between our approach and randomized smoothing on CIFAR-10 dataset (reviewers d51P and TteL).
- [Appendix E] We design complete ablation experiments to investigate our training method in detail (reviewers THnD and WMpC).
- [Section 5.3 and Appendix F] We provide sensitivity analysis over hyper-parameters $\theta$ and $\lambda$. We show our approach is not sensitive to these hyper-parameters (reviewer THnD).

**Reproducibility**:

We have uploaded the code and provided detailed commands as the reviewer suggested (reviewer THnD).

We notice that the discussion phase transition is approaching and currently we have not received any feedback. We sincerely hope our response can address the concerns raised by the reviewers. We would be grateful if the reviewers can re-evaluate our paper based on the revised version of the paper and our response. Please let us know if any questions or concerns remain. Thank you.

Sincerely,

Paper 3247 Authors

---

### Decision · Program_Chairs · 2022-01-20

**Decision:**

Accept (Poster)

**Comment:**

This paper is a follow-up paper of Zhang et al. (2021), that proposed a new network architecture for adversarial robustness, l_\infty distance net. Although the l_\infty network is provably 1-Lipschitz w.r.t. the l_\infty distance, its training procedure exploits the l_p relaxation to overcome the non-smoothness of the model but suffers from an unexpected large Lipschitz constant at the early training stage, an issue to be solved. This paper resolves this issue by a new loss design of scaled cross entropy loss and clipped hinge loss. Without using MLP on top of the l_\infty distance net backbone, the proposed new training method empirically outperforms the original one in Zhang et al. (2021) and improves over the state-of-the-art by more than 5% for 8/255 and other radiuses. Moreover, the paper shows the theoretical expressive power of l_\infty distance net for well-separated data.

There are some concerns about the moderate novelty and reproducibility of the results. Since the empirical results are indeed impressive, the paper could be accepted conditional on that the authors release their reproducible codes to the public.